# Pulmonary mesenchymal stem cells are engaged in distinct steps of host response to respiratory syncytial virus infection

Melanie Brügger[1,2,3], Thomas Démoulins[1,2☯], G. Tuba Barut[1,2,3☯],
Beatrice Zumkehr[1,2], Blandina I. Oliveira Esteves[1,2], Kemal Mehinagic[1,2,3],
Quentin Haas[4], Aline Schögler[5], Marie-Anne Rameix-Welti[6], Jean-François Eléouët[7],
Ueli Moehrlen[8], Thomas M. Marti[5,9], Ralph A. Schmid[5,9], Artur Summerfield[1,2],
Horst Posthaus[2,10], Nicolas Ruggli[1,2], Sean R. R. Hall[9,11], Marco P. Alves[1,2]*

1 Institute of Virology and Immunology, University of Bern, Bern, Switzerland, 2 Department of Infectious Diseases and Pathobiology, Vetsuisse Faculty, University of Bern, Bern, Switzerland, 3 Graduate School for Cellular and Biomedical Sciences, University of Bern, Bern, Switzerland, 4 Institute of Pharmacology, University of Bern, Bern, Switzerland, 5 Department of Biomedical Research, University of Bern, Bern, Switzerland, 6 Université Paris-Saclay, INSERM, Université de Versailles St. Quentin, UMR 1173 (2I), Versailles, France, 7 Université Paris-Saclay, INRAE, UVSQ, VIM, Jouy-en-Josas, France, 8 Pediatric Surgery, University Children's Hospital Zurich, Zurich, Switzerland, 9 Department of General Thoracic Surgery, Inselspital, Bern University Hospital, University of Bern, Bern, Switzerland, 10 Institute of Animal Pathology, Vetsuisse Faculty, University of Bern, Bern, Switzerland, 11 Gillies McIndoe Research Institute, Wellington, New Zealand

☯ These authors contributed equally to this work.
* marco.alves@vetsuisse.unibe.ch

**Data Availability Statement:** All relevant data are within the manuscript and its Supporting Information files.

## Abstract

Lung-resident (LR) mesenchymal stem and stromal cells (MSCs) are key elements of the alveolar niche and fundamental regulators of homeostasis and regeneration. We interrogated their function during virus-induced lung injury using the highly prevalent respiratory syncytial virus (RSV) which causes severe outcomes in infants. We applied complementary approaches with primary pediatric LR-MSCs and a state-of-the-art model of human RSV infection in lamb. Remarkably, RSV-infection of pediatric LR-MSCs led to a robust activation, characterized by a strong antiviral and pro-inflammatory phenotype combined with mediators related to T cell function. In line with this, following *in vivo* infection, RSV invades and activates LR-MSCs, resulting in the expansion of the pulmonary MSC pool. Moreover, the global transcriptional response of LR-MSCs appears to follow RSV disease, switching from an early antiviral signature to repair mechanisms including differentiation, tissue remodeling, and angiogenesis. These findings demonstrate the involvement of LR-MSCs during virus-mediated acute lung injury and may have therapeutic implications.

## Author summary

This work identifies a novel function of lung-resident MSCs during virus-induced acute lung injury. These findings contribute to the understanding of host response and lung

**Funding:** This work acknowledges support from the Swiss National Science Foundation to MPA (www.snf.ch, project 310030_172895), that included partial salary support for MB and TD. This work was also supported by a grant from the Gottfried and Julia Bangerter-Rhyner Foundation to MPA (www.bangerter-stiftung.ch), that included partial salary support to MB. The funders had no role in study design, data collection and analysis, decision to publish, or preparation of the manuscript.

**Competing interests:** The authors have declared that no competing interests exist.

repair mechanisms during a highly prevalent clinical situation and may have therapeutic implications.

## Introduction

The ability of organs to maintain homeostasis and regenerate following injury is vital to an organism. These processes are maintained by many supportive cells including tissue-resident mesenchymal stem and stromal cells (MSCs). MSCs are found in nearly every vascularized tissue including the upper and lower respiratory tract. More specifically, they are localized in perivascular niches of small and larger blood vessels and were shown to be lung-resident [1–7]. In the alveolar niche, lung-resident (LR)-MSCs can interact with epithelial cells and promote alveolar cell growth, differentiation, and self-renewal. This is of particular importance for epithelial maintenance as well as for repair and regeneration as demonstrated in artificial rodent models of lung injury [8–10]. Failure of these mechanisms may play a role in the etiology of several chronic lung diseases [11,12]. LR-MSCs can directly interact with various pulmonary immune cell populations via cellular contact or in a paracrine manner by secretion of soluble factors [13]. Characterization of immunomodulatory properties of mesenchymal cells are of great clinical importance since non-resident MSCs are the subject of cell-based treatment approaches for various lung disorders [14,15].

The human respiratory syncytial virus (RSV) is of high prevalence and causes a huge burden on public health systems. Globally, RSV is the leading cause of severe acute lower respiratory tract infections in early childhood and suspected to have an underestimated impact on the elderly [16,17]. Several animal models have been established to study RSV infection, with the lamb model best reflecting anatomical and immunological properties of the neonatal human lung. This model recapitulates the clinical features of the human pediatric disease and the closely related bovine RSV is a natural pathogen of ruminants. Furthermore, unlike rodents, lambs present a similar lung development, are susceptible to human RSV, and develop comparable pulmonary lesions [18–20].

Despite intense research on MSCs, no information is available about their role during acute respiratory infection and their contribution following virus-induced injury. Given the important role of the mesenchymal compartment and LR-MSCs in lung regeneration and repair, we hypothesize that this applies during respiratory virus infection. Therefore, we aimed to explore the role of the pulmonary mesenchymal compartment, in particular LR-MSCs, during RSV infection using an experimental approach based on primary LR-MSCs isolated from human pediatric donors and a translational model of human RSV infection in lamb.

## Results

### LR-MSCs are highly permissive to RSV infection

To study the immunobiology of LR-MSCs during respiratory virus infection, we determined first if these cells are susceptible towards RSV infection in comparison to a well described cellular target of RSV, namely airway epithelial cells (AECs) [21–23]. We characterized the LR-MSCs isolated from pediatric donors according to the three criteria proposed by the Mesenchymal and Tissue Stem Cell Committee of the International Society for Cellular Therapy [24]. As shown in Fig 1A, LR-MSCs were positive for the three MSC surface markers CD73, CD90, and CD105. Additionally, we confirmed plastic adherence of the cells and their trilineage potential by differentiating them towards chondrocytes, osteocytes, and adipocytes (Fig

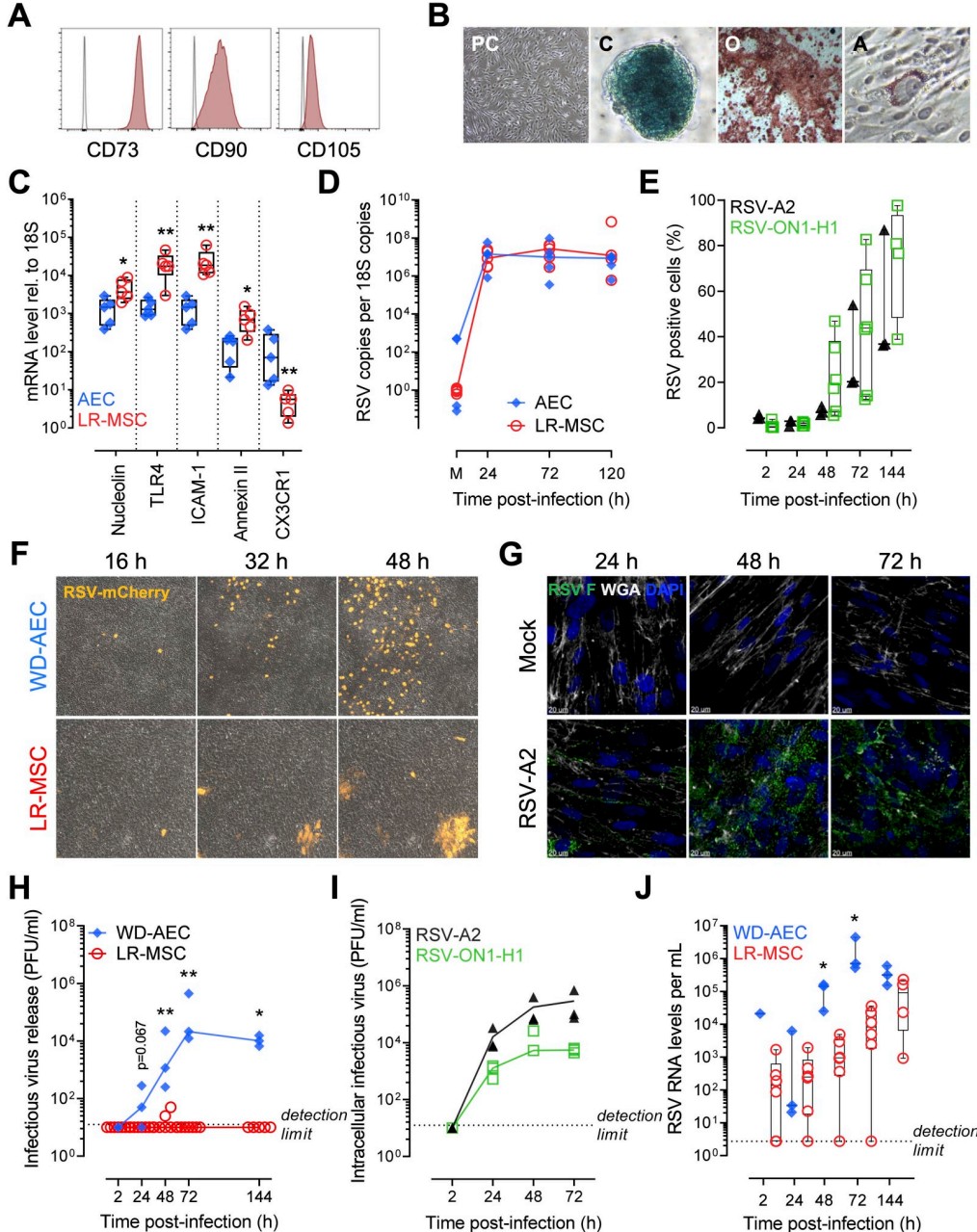

**Fig 1. LR-MSCs are highly permissive to RSV infection.** (A) Representative histograms showing expression of the surface markers CD73, CD90, CD105 on pediatric LR-MSCs. (B) Representative phase-contrast (PC) micrograph showing morphology in culture and demonstrates plastic adherence. Representative images of Toluidine blue, Alizarin Red S, and Oil Red O stainings after chondrogenic (C), osteogenic (O), and adipogenic (A) differentiation, respectively. Magnification 40X (PC, O) and 200X (C, A). (C) mRNA expression levels of RSV receptors relative to $10^6$ 18S. A Mann-Whitney U test was applied to compare the two cell types (AECs *versus* LR-MSCs). Boxplots indicate the median value (centerline) and interquartile ranges (box edges), with whiskers extending to the lowest and the highest values. Each symbol represents an individual donor (n = 5). *p<0.05, **p<0.01. (D) Intracellular viral loads in AECs and LR-MSCs over time following infection with RSV-A2 at a MOI of 1 PFU/cell expressed as RSV copies per $10^{12}$ 18S copies. Each symbol represents an individual donor (n = 5). (E) RSV F-protein positive LR-MSCs assessed by FCM and plotted over time. LR-MSCs were infected with 0.1 PFU/cell with RSV-A2 (n = 3) or a clinical isolate RSV-ON1-H1 (n = 4–6). Each symbol represents an individual donor. (F) Representative live-cell imaging of WD-AECs and LR-MSCs infected with 0.1–0.5 PFU/cell with RSV-mCherry and followed over time. The micrographs were taken at 16, 32, and 48 h p.i. in the same area of the cellular layer. (G) Representative confocal microscopy evaluation of LR-MSCs noninfected (mock) or infected with RSV-A2 at 1 PFU/cell 24 to 72 hours p.i. RSV, green; DAPI, dark blue; WGA, white. Scale bar, 20 μm. (H)

Supernatants of infected LR-MSCs or apical washes of infected WD-AEC cultures were analyzed by a PFU assay. Cells were infected with RSV-ON1-H1 at a MOI of 0.1 PFU/cell. A Mann-Whitney U test was applied to compare the two cell types (WD-AECs, n = 3 *versus* LR-MSCs, n = 5–6). Each symbol represents an individual donor. *p<0.05, **p<0.01. (I) Intracellular infectious RSV titers in LR-MSCs infected with RSV-A2 or RSV-ON1-H1 at 0.1 PFU/cell. Each symbol represents an individual donor (n = 3) (J). Extracellular RSV RNA load over time in supernatants of infected LR-MSCs or apical washes of infected WD-AEC cultures. Cells were infected with RSV-ON1-H1 at a MOI of 0.1 PFU/cell. A Mann-Whitney U test was applied to compare the two cell types (WD-AECs, n = 3 *versus* LR-MSCs, n = 4–6). Each symbol represents an individual donor. *p<0.05.

1B). Next, we assessed mRNA expression levels of different putative RSV cell-surface receptors/co-receptors [25–30], namely nucleolin, toll-like receptor 4 (TLR4), intercellular adhesion molecule 1 (ICAM-1), annexin II, and CX3CR1 in pediatric LR-MSCs in comparison to donor-matched AECs. All tested receptors were expressed in both cell types at various levels (Fig 1C), suggesting that LR-MSCs are susceptible to RSV infection. Thereby, we detected fast replication kinetics of RSV RNA in LR-MSCs similar to levels measured in infected AECs (Fig 1D). To confirm the replication of RSV in LR-MSC, we next used a flow cytometry (FCM) approach. We infected LR-MSCs with a clinical isolate of RSV subtype A (RSV-ON1-H1) and with RSV-A2 at low multiplicity of infection (MOI; 0.1 PFU/cell). After 144 hours post-infection (p.i.) 40–100% of LR-MSCs were infected depending on the donor (Fig 1E). However, when using higher MOIs (1 PFU/cell), 144 hours p.i., both RSV-A2 and RSV-ON1-H1 infected nearly 100% of the cells (S1A Fig). Given that LR-MSCs are highly susceptible to RSV infection, we were wondering how the virus spreads among the cells. To follow visually virus spread in the two different cell types, we performed live imaging in specific areas of the cellular layer following infection with a recombinant RSV construct expressing constitutively the mCherry reporter (RSV-mCherry). While infected well-differentiated (WD)-AECs appear as discrete RSV-positive cells, infected LR-MSCs showed up in discrete foci, suggesting RSV spread via cell-to-cell contact (Fig 1F). In order to confirm that the increase of mCherry reporter signal in LR-MSCs foci is due to RSV spread, we infected LR-MSCs with RSV-A2 and visualized the presence of RSV over time using a high definition confocal microscopy approach. Notably, while the RSV signal was often located at the plasma membrane at 24h p.i, it increased over time and became mainly perinuclear and/or cytosolic at later time points, suggesting replication (Fig 1G). To further confirm the distinct life cycles, we measured infectious virus release over time in LR-MSCs in comparison to WD-AECs by applying a plaque-forming unit (PFU) assay. Remarkably, at most of the time-points tested, there was significantly higher infectious virus release in the apical washes of infected WD-AECs compared to the supernatants of infected LR-MSC cultures. Moreover, while there was almost no infectious virus detectable in the supernatants of LR-MSCs with both MOIs tested (Figs 1H and S1B), we observed an exponential increase of the intracellular infectious RSV titers over time at both MOIs tested (Figs 1I and S1C). Finally, when assessing the extracellular viral RNA loads in WD-AECs in comparison to LR-MSCs, we observed a rapid exponential increase of virus loads in the apical washes of WD-AECs and rather an accumulation of RSV RNA in the supernatants of LR-MSCs, suggesting the presence of substantial levels of non-infectious RSV particles (Figs 1J and S1D). Altogether, these results demonstrate that primary pediatric LR-MSCs are highly permissive to RSV infection and that RSV replicates in LR-MSC, although the life cycle of RSV is distinct to AECs.

## RSV infection is altering the immune properties of LR-MSCs

We aimed to establish if LR-MSCs can mount an antiviral response upon RSV infection. Twenty-four hours p.i., we observed a significant increased expression of interferon stimulated

genes (ISGs) commonly induced during RNA virus infections [31]. More specifically, RSV-A2 and RSV-ON1-H1 infection induced a significant upregulation of retinoic acid-inducible gene-I (RIG-I), melanoma differentiation-associated protein 5 (MDA-5), 2',5'-oligoadenylate synthetase 2 (2',5'-OAS), interferon (IFN)-induced dynamin-like GTPase (MxA), and virus inhibitory protein, endoplasmic reticulum-associated, IFN-inducible (Viperin) compared to mock control (Fig 2A). Furthermore, the mRNA levels of both IFN-β and IFN-λ1 had a trend towards significantly higher levels upon RSV infection (Fig 2B). We also verified if RSV infection-induced mRNA levels of type I and III IFNs resulted in an increased protein release. IFN-β release was significantly stimulated by the two RSV strains over time and was quantified at highly significant levels after 72 hours p.i. (Fig 2C). Furthermore, the protein levels of IFN-λ1/ 3 were significantly elevated after 24 and 72 hours p.i. compared to mock control for RSV-A2 and had a trend towards significance for the RSV-ON1-H1 strain (Fig 2D). Notably, infection of WD-AECs in comparison to LR-MSCs induced comparable IFN type I and III levels 24 to 72 hours p.i., suggesting a similar IFN response upon RSV infection (S2A, S2B, and S2C Fig). Next, we wanted to evaluate whether infection has an impact on the immunomodulatory properties of LR-MSCs. First, we evaluated the cell-surface expression levels of two co-stimulatory molecules, namely programmed death-ligand 1 (PD-L1) and major histocompatibility complex (MHC) class I, described in LR-MSCs to be involved in their immunomodulatory characteristics [32,33]. LR-MSCs expressed baseline levels of both co-stimulators. Poly(I:C), a ligand mimicking RNA virus infection, induced a significant upregulation of both co-stimulatory molecules in comparison to mock control. In line with this, RSV infection led to the induction of slightly increased levels of PD-L1 and MHC class I expression compared to mock-treated LR-MSCs, with differences not reaching significance, as shown in representative histograms (S2D Fig) and upon quantification (S2E Fig). Next, we evaluated the secretory profiles of LR-MSCs following RSV infection in comparison to WD-AECs. Upon infection of WD-AECs with RSV-A2 or RSV-ON1-H1, the basolateral cytokine secretion revealed mainly CXCL10/ IP-10 and CXCL8/IL-8 release (Fig 2E). Contrary to WD-AECs, RSV infection of LR-MSCs led to a massive secretion of several cytokines. In fact, 24 and 72 hours p.i., expression was remarkably increased for most of the cytokines measured, including CXCL10/IP-10, CXCL8/ IL-8, IL-6, G-CSF, CCL5/RANTES, IL-9, IFN-γ, IL-5, and L-17 to name the most notable ones (Fig 2F). Taken together, these data demonstrate that RSV infection leads to a robust activation of LR-MSCs, characterized by a strong antiviral and pro-inflammatory phenotype combined with cytokines modulating T cell function (Fig 2G).

## RSV infection is causing lung injury in lamb

To study the MSC compartment of the lung and its role during an acute virus infection, we applied an *in vivo* approach using the neonatal lamb model of RSV infection. Although data are available on ovine MSCs derived from several tissues, ovine LR-MSCs have not been described yet [34–36]. Thus, we isolated and expanded LR-MSCs cultures from healthy ovine lung cell suspensions as described previously for human tissue [3]. Characterization by immunophenotyping displayed a high level of cell surface markers typically found in MSCs such as the hyaluronate receptor CD44 and CD29 (ITGB1). An additional marker, CD166 (ALCAM), was also expressed on cultured MSCs (S3A Fig). The plastic-adherent cells were large with a fibroblast-like morphology. Multilineage capacity was confirmed following specific culture conditions, as ovine LR-MSCs transdifferentiated to chondrocytes, osteocytes, and adipocytes (S3B Fig). Together, these features fulfill the accepted criteria to identify MSCs proposed by the Mesenchymal and Tissue Stem Cell Committee of the International Society for Cellular Therapy [24]. Before infection of the animals, we performed an infection of ovine LR-MSC

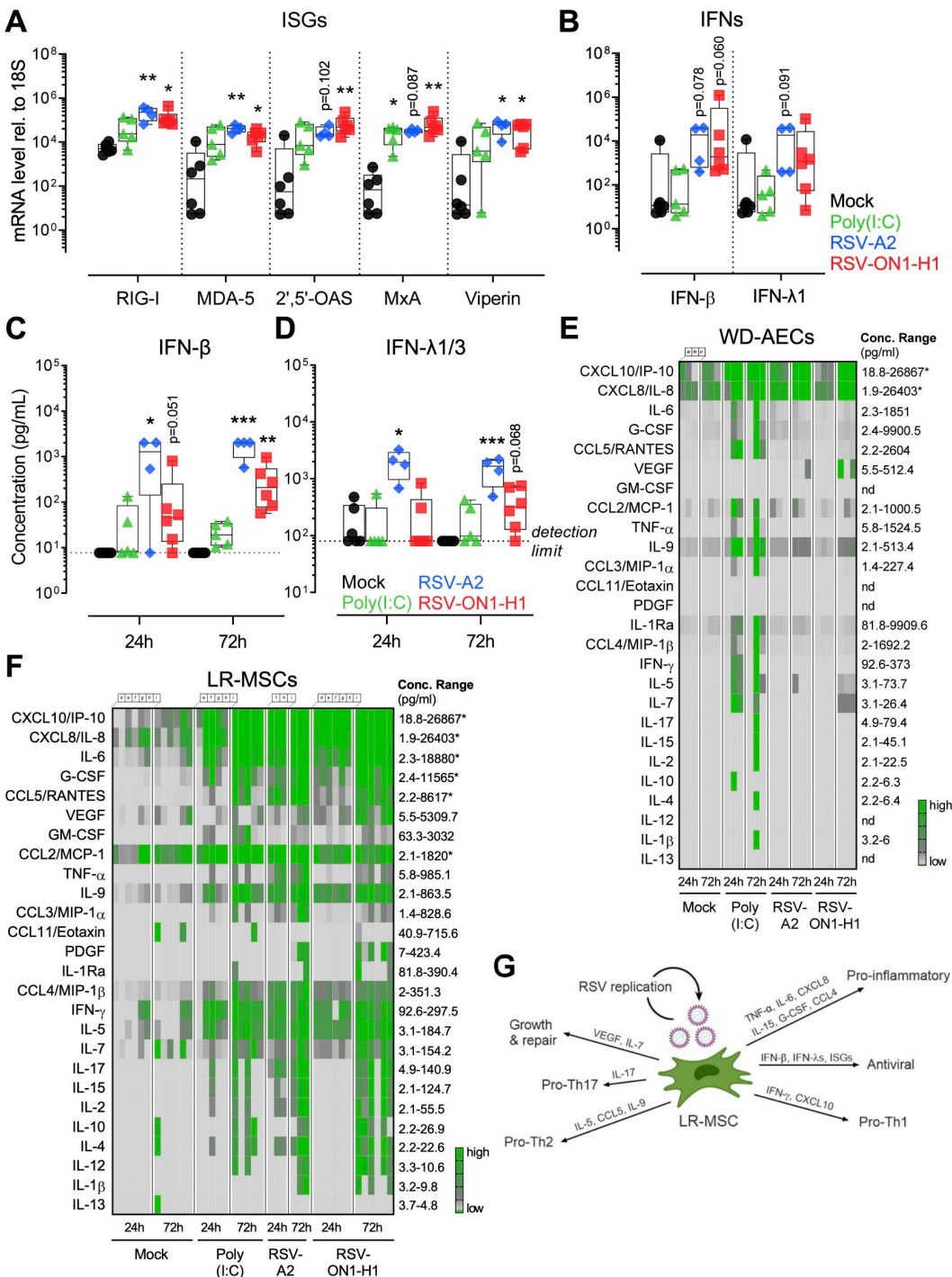

**Fig 2. RSV infection is altering the immune properties of LR-MSCs.** (A, B) mRNA levels of selected ISGs (A) and IFNs (B) in LR-MSCs treated with mock control, poly(I:C) 10 μg/ml, RSV-A2, or RSV-ON1-H1 for 24 h at 1 PFU/cell. Boxplots indicate the median value (centerline) and interquartile ranges (box edges), with whiskers extending to the lowest and the highest values. Each symbol represents an individual donor (mock, n = 6; poly(I:C), n = 5; RSV-A2, n = 4; RSV-ON1-H1, n = 6). The data were compared with the Kruskal–Wallis test followed by Dunn's post hoc test. *p<0.05, **p<0.01. (C, D) IFN-β (C) and IFN-λ1/3 (D) protein levels in supernatants of LR-MSCs treated with mock control, poly(I:C) 10 μg/ml, RSV-A2, or RSV-ON1-H1 for 24 h and 72 h at 1 PFU/cell. Boxplots indicate median value (centerline) and interquartile ranges (box edges), with whiskers extending to the lowest and the highest values. Each symbol represents an individual donor (mock, n = 6; poly(I:C), n = 5; RSV-A2, n = 4; RSV-ON1-H1, n = 6). A Kruskal–Wallis test followed by a Dunn's post hoc test was applied to compare the different groups. *p<0.05, **p<0.01, ***p<0.001. The detection limits of the assays are

indicated with the dotted line at 7.7 pg/mL and 79.8 pg/mL for IFN-β and IFN-λ1/3, respectively. (E, F) Multiplex assay of basolateral medium of WD-AECs (E) or supernatants of LR-MSCs (F), 24 and 72h after treatment with mock control, 10 μg/ml poly(I:C), RSV-A2, or RSV-ON1-H1 for 24h and 72h at 1 PFU/cell. The concentration ranges of the different cytokines are indicated (lower detection limit-highest concentration measured). An asterisk is present when the concentration of the sample was higher than the upper range of the assay. Each column represents a different donor (mock, n = 6; poly(I:C), n = 5; RSV-A2, n = 4; RSV-ON1-H1, n = 6; donors a-c for WD-AECs and d-i for LR-MSCs). (G) Overview of the impact of RSV infection on LR-MSCs. RSV-mediated activation of LR-MSCs is characterized by an antiviral and pro-inflammatory phenotype combined with cytokines promoting T helper cell (Th) polarization.

cultures with human RSV. Virus replication was evident as observed by an increase of RSV load of cell-associated viral RNA over time. Similarly to RSV-infected human LR-MSCs, the infectious virus in the supernatants was undetectable or close to the detection limit suggesting cell-to-cell spread (S3C Fig). Furthermore, *ex vivo* infection of ovine precision-cut lung slice (PCLS) cultures with a recombinant RSV construct expressing constitutively the green fluorescent protein (RSV-GFP) led to an increase of the reporter signal over time, indicating replication (S3D Fig). When analyzing the infected PCLSs at higher magnification, the GFP signal was mainly located in the alveolar wall suggesting infection of pneumocytes (S3E Fig). Newborn lambs were infected with human RSV-A2 and respiratory disease was evaluated during the acute (3 and 6 days p.i.), recovery (14 days p.i.), and convalescence (42 days p.i.) phases. To do so, at each timepoint p.i., we examined the lung tissue and the cellular components of bronchoalveolar lavage (BAL) fluid (Fig 3A). RSV-infected neonates showed clear nasal discharge, occasional coughing and wheezing. Macroscopic lesions were detected at necropsies performed 3 and 6 days p.i. Lungs failed to collapse and showed multiple, irregular, mildly reddened areas with slightly increased consistency of up to 2 cm in diameter in all lung lobes. Histopathological evaluation of the lungs revealed tissue consolidation, bronchiolitis, and thickening of interalveolar walls 3, and 6 days following RSV infection (Fig 3B). Following infection, alveolar type 2 cell hyperplasia was observed, indicating tissue injury (S4A Fig). In addition, at 6 days p.i. we observed occasionally potential syncytial cells lining alveoli (S4A and S4B Fig). These features disappeared 14 days p.i. (Fig 3B), indicating resolution of lesions after RSV-mediated acute lung injury. Immunohistochemistry analysis of lung tissue sections revealed the presence of RSV antigen in pneumocytes (Fig 3C). We evaluated the viral RNA levels in the lung and BAL tissues. In the lung, the RSV loads decreased over time with no virus detectable in 50% of the animals 14 days p.i. and in all animals 42 days p.i. In the cellular fraction of the BAL, RSV copies were highest at 3 days p.i., with viral RNA detectable in 66% of the animals 14 days p.i. and reaching undetectable levels 42 days p.i. There was a significant difference in viral loads with approximately 100 times higher level of RSV copies detectable in the BAL compartment compared to the lung at both, 3 and 6 days p.i. (Fig 3D). To confirm the histopathological evaluation revealing RSV-mediated tissue injury, we assessed cell death by quantifying cleaved caspase-3-positive cells using an approach summarized in S5 Fig. The percentage of cleaved caspase-3 positive cells in the lung, was significantly increased for infected animals 3, and 6 days p.i. and similar 14, and 42 days p.i. compared to mock controls (Fig 3E). Lung injury is further indicated by shedding of apoptotic cells into the bronchoalveolar space. Indeed, there are significantly more cleaved caspase-3 positive cells 3 days p.i. assessed for infected compared to mock-infected animals (Fig 3F). The impact of RSV infection on the bronchoalveolar space is further indicated by an infiltration of cells as assessed in the BALs from infected compared to mock-infected animals 3 to 14 days p.i. (Fig 3G). In summary, the life cycle of RSV is similar in human and ovine LR-MSCs and infection of neonatal lambs with human RSV is causing lung injury. These data support the use of the lamb model to investigate LR-MSCs in RSV disease.

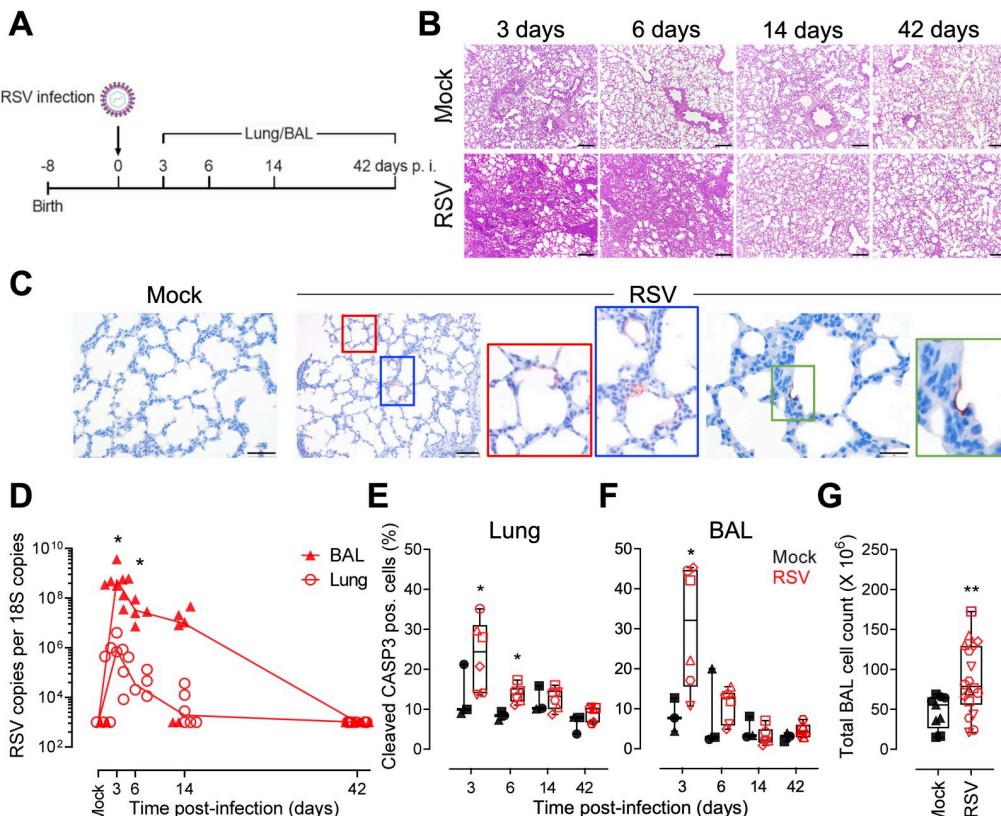

**Fig 3. RSV infection is causing lung injury in lamb.** (A) Newborn animals were trans-tracheal inoculated with $10^8$ PFU per animal of the human strain RSV-A2 or PBS (mock). Animals were euthanized 3, 6, 14, and 42 days p.i. and lung and BAL tissues were harvested for histopathological evaluation, qPCR, and FCM analysis (B) Representative H&E stained histopathological sections of the lung tissue from noninfected (mock) and RSV-infected lambs at 3, 6, 14, and 42 days p.i. Scale bar, 200 μm. (C) Histological lung sections from animals 6 days p.i. stained for RSV (red) and counterstained with haematoxylin (blue). From left to right panels, scale bars 50 μm, 100 μm and 20 μm, respectively. (D) Viral load in lungs and the cellular fraction of the BAL of infected lambs measured 3, 6, 14, and 42 days p.i. Comparison of viral loads between BAL and lung tissues was done with a one-way ANOVA and the Tukey post-hoc test. Each symbol represents an individual animal (n = 6 per timepoint). *$p < 0.05$. (E, F) Frequency of cleaved caspase 3 (CASP3)-positive cells in the lung (E) and BAL (F) of infected animals 3 to 42 days p.i. Boxplots indicate the median value (centerline) and interquartile ranges (box edges), with whiskers extending to the lowest and the highest values. Each symbol represents an individual animal (mock, n = 3 and RSV, n = 6). Multiple comparison was done with a one-way ANOVA and the Tukey post-hoc test. *$p < 0.05$. (G) Total BAL cell counts in mock and RSV-infected animals 3, 6, and 14 days p.i. pooled. Boxplots indicate the median value (centerline) and interquartile ranges (box edges), with whiskers extending to the lowest and the highest values. Each symbol represents an individual animal (n = 9, mock and n = 18, RSV). A Mann-Whitney U test was applied to compare the two groups. **$p < 0.01$.

## LR-MSCs are a target for RSV infection *in vivo*

To determine if LR-MSCs can be a target for RSV *in vivo*, we designed an FCM assay allowing the identification of the pulmonary epithelial (CD31⁻CD45⁻panCTK⁺) and mesenchymal (CD31⁻CD45⁻panCTK⁻CD29⁺CD44⁺) compartments, as well as detection of RSV infection *in vivo* (Fig 4A). In 13–80% of the infected animals, both RSV-positive epithelial cells and LR-MSCs were detected in the lung cell suspensions 3, 6 and 14 days p.i. (Figs 4B and S6). When RSV-positive LR-MSCs were plotted as a function of RSV-positive epithelial cells, a significant positive association was found, suggesting that the spread of RSV infection to LR-MSCs is linked to the extent of replication in the pulmonary epithelium (Fig 4C). Next, MSCs derived from lung tissue and BALs, were isolated from each animal at the different time

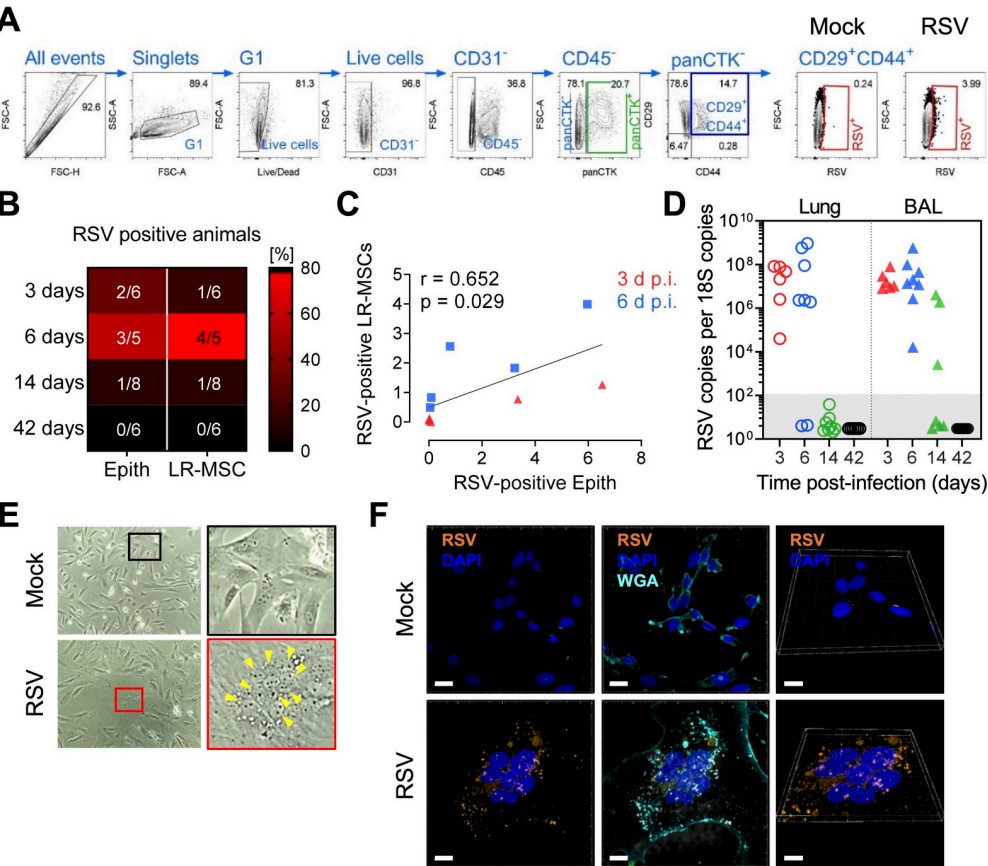

**Fig 4. LR-MSCs are a target for RSV infection *in vivo*.** (A) FCM gating strategy to identify RSV-infected epithelial cells (CD31⁻CD45⁻panCTK⁺) and LR-MSCs (CD31⁻CD45⁻panCTK⁻CD29⁺CD44⁺). panCTK, pan-cytokeratin; G1, gate 1. (B) Percentage of animals with RSV-positive epithelial and LR-MSCs 3 to 42 days p.i. The fractions indicate the number of infected animals where infection was detected by FCM compared to the total number of animals. (C) Correlation of RSV-positive LR-MSCs and RSV-positive epithelial cells at 3 and 6 days p.i. Associations were tested using the Spearman rank correlation test. Each symbol represents an individual animal. (D) RSV loads in MSCs isolated from lung and BAL tissues of RSV-infected animals 3, 6, 14, and 42 d p.i. and expanded in culture. Each symbol represents an individual animal (n = 6–8). The filled dashed box indicates the samples below detection limit. (E) Phase-contrast micrographs of LR-MSCs derived from BALs expanded in culture from noninfected (mock) or infected (RSV) animals 3 days p.i. Magnification 100X. The yellow arrowheads indicate a cluster of nuclei of a potential syncytium. (F) Representative confocal microscopy evaluation of LR-MSCs derived from BALs expanded in culture from noninfected (mock) or infected (RSV) animals 3 days p.i. RSV, orange; DAPI, dark blue; WGA, light blue. Scale bars, 15 μm (left and middle panels) and 10 μm for the 3D capture (right panels).

points p.i. and expanded in culture. When testing the presence of RSV within the lung-derived MSC cultures, viral RNA was detectable in most cultures from the animals at 3 and 6 days p.i. and in none of the cultures isolated from animals at 14 and 42 days p.i. In line with the higher viral loads in the BAL cellular fraction compared to the lung tissue, we detected high levels of RSV RNA in all BAL-derived MSCs from animals at 3 and 6 days p.i. and from 3 out of 7 animals isolated at 14 days p.i. RSV RNA levels were undetectable in all cultures isolated from animals at 42 days p.i. (Fig 4D). Interestingly, we noticed giant multinucleated cells in cultures derived from infected animals which were never seen for mock animal-derived cultures (Fig 4E). Given that RSV spread in foci and infectious virus was rarely detected in the supernatants of infected human and ovine LR-MSCs cultures, we hypothesized that these phenotypically distinct cells were RSV-infected MSCs. This was confirmed by confocal microscopy

visualization (Fig 4F) and by the infection of cultures from human and ovine LR-MSCs with RSV-mCherry (S7A and S7B Fig). Together, these results indicate that the lung MSC compartment is a target for RSV infection *in vivo* during the early phase of respiratory disease.

## RSV infection leads to the expansion of the pulmonary MSC niche *in vivo*

To investigate the impact of virus-mediate lung injury on LR-MSCs *in vivo*, we analyzed the mesenchymal compartment of RSV-infected newborn lambs during the acute, recovery, and convalescence phases of RSV disease. As a mild to asymptomatic RSV disease control, adult animals were infected for a period of 14 days (Fig 5A). Contrary to RSV-infected neonates, infection of adults did not lead to any notable clinical manifestation nor macroscopic lesions, while rare foci of interstitial thickening with leukocyte infiltrates and mild alveolar type 2 hyperplasia were visible histologically (Fig 5B). A colony-forming unit-fibroblast (CFU-F) assay was applied to lung-derived cells from infected and mock-infected animals. This assay is commonly used to assess the proliferative activity of MSCs and their ability to form discrete fibroblast-like colonies [37,38]. The analysis revealed a significant effect of RSV infection on the proliferative properties of LR-MSCs by increasing their activity already 3 days p.i. compared to mock control (Fig 5C). Quantitative analysis confirmed significantly increased CFU-F counts during the acute phase of infection 3 and 6 days p.i. Interestingly, there was still a nonsignificant tendency of more CFU-F counts during later phases of RSV disease 14 and 42 days p.i. compared to age-matched non-infected animals (Fig 5D). CFU-F counts were around

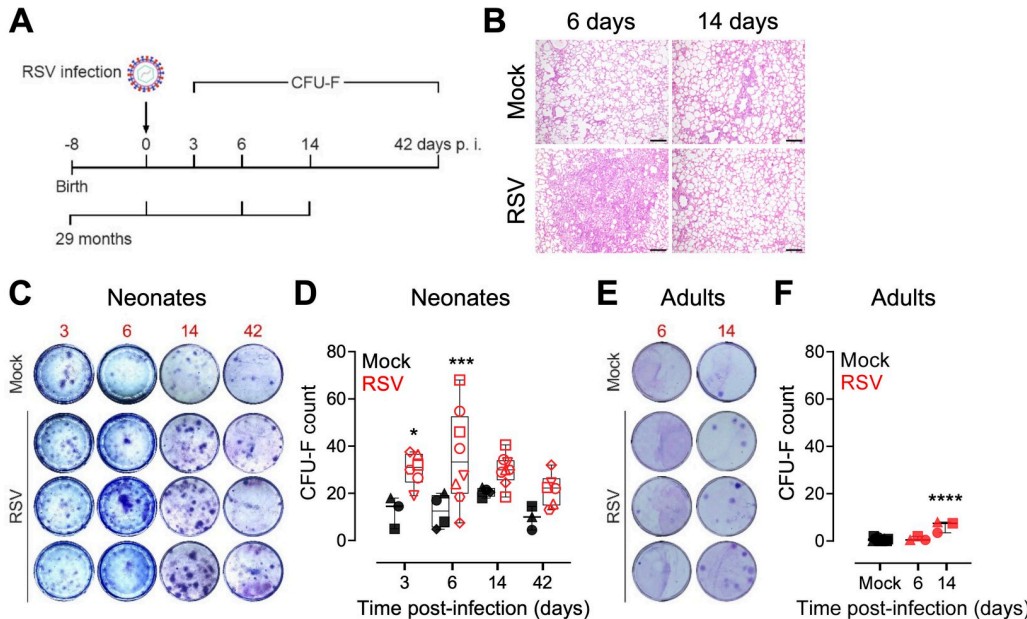

**Fig 5. RSV infection leads to the expansion of the pulmonary MSC niche *in vivo*.** (A) Newborn or adult (average age of 29 months) animals were trans-tracheal inoculated with $10^8$ PFU per animal of the human strain RSV-A2 or PBS (mock). Newborns were euthanized 3, 6, 14, and 42 days p.i. and adults 6 and 14 days p.i. and lung tissue was harvested for CFU-F assay. Lung tissue of adults was harvested for histopathological evaluation (B) Representative H&E stained histopathological sections of the lung tissue from noninfected (mock) and RSV-infected adults at 6 and 14 days p.i. scale bar, 200 μm. (C, E) Representative images of the CFU-F assay for mock control and RSV-infected neonates (C) and adults (E) 3 to 42 days p.i. and 6 and 14 days p.i., respectively. Each image represents an individual animal. (D, F) Quantification of the CFU-F assay for each neonate (D) and adult (F) animals, given as CFU-F count relative to $3.33 \times 10^5$ nucleated cells over time. Boxplots indicate the median value (centerline) and interquartile ranges (box edges), with whiskers extending to the lowest and the highest values. Each symbol represents an individual animal (neonates, per timepoint: mock, n = 3–4; RSV, n = 6–8; adults: mock, n = 14; RSV, n = 3 per timepoint). A one-way ANOVA and the Holm-Sydak post-hoc test was applied to compare differences between groups. *p<0.05, ***p<0.001, ****p<0.0001.

20 times lower for control adults (average CFU-F count of 0.6) in comparison to control neonates (average CFU-F count of 14.2), indicating an age-dependent effect on the steady-state proliferative properties of LR-MSCs. However, RSV infection led to a significant increase of these colonies 14 days p.i. with an average count of 6.3 (average CFU-F count of 29.9 for infected neonates 14 days p.i.) (Fig 5E and 5F). These results provide evidence that RSV infection leads to a stimulation of the proliferative activity and the expansion of the pulmonary MSC pool, particularly in neonates compared to older animals.

## The transcriptional activity of LR-MSCs is related to the phases of RSV disease

To investigate how RSV infection impacts LR-MSCs at the transcriptional level *in vivo*, we applied an experimental approach developed by Spadafora *et al.*, who showed that tracheal aspirate-derived MSCs retain their transcriptional signature following a short expansion in culture [39]. We undertook whole transcriptome analysis from expanded LR-MSCs isolated from infected neonates during the acute, recovery and convalescent phases of RSV disease in comparison to mock-infected neonates. To confirm identity of MSCs expanded in culture, we calculated the transcripts per million (TPM) of markers used for their identification [24,40]. LR-MSCs had high transcriptional levels of ENG (CD105), NT5E (CD73), THY1 (CD90), ITGB1 (CD29), CD44 and PDGFRA among others, and lacked transcripts of HLA-DR or HLA-DQ, PTPRC (CD45), CD34, ITGAM (CD11b), or PECAM (CD31) confirming their MSC identity (Fig 6A). Next, we compared the transcriptome profiles of LR-MSCs during the acute phase of RSV disease (6 days p.i.) to mock-infected animal-derived LR-MSCs and found 14 differentially expressed genes (DEGs) to be significantly upregulated. The majority of them belong to the IFN pathway and are associated with antiviral responses, such as RSAD2 (Viperin), OASL, ISG15, IFI44, and IFI44L and two pro-apoptotic genes, namely TNFSF10/TRAIL and STC1 (Fig 6B). These findings are in line with our *in vitro* results derived from human LR-MSC and support the idea that LR-MSCs mount an antiviral response against RSV infection *in vivo*. Remarkably, while there was no significant difference in the recovery phase (14 days p.i.) compared to mock-infected control (Fig 6C), there were 43 DEGs in LR-MSCs isolated during the convalescence phase (42 days p.i.) (Fig 6D). Several of these genes are associated with the secretory pathway, including MAGE2 (protein trafficking), TBC1D20 (autophagosome maturation), MAP7D3 (microtubule assembly), and MARCHF9 (protein processing). In line with this, high transcriptional activity of LR-MSCs during the convalescence phase is evidenced by the upregulation of various non-coding RNAs and ribonuclease proteins involved in pre-RNA processing and regulation of transcription and/or translation. Furthermore, genes linked with differentiation and developmental processes, such as SOX4, NOC3L, and C2CD3 were differentially expressed during the convalescence phase. Remarkably, many of the upregulated DEGs were related to endothelial cell function and angiogenesis, such as TRPC4 or KDR which codes for the vascular endothelial growth factor receptor 2 (VEGFR2), and ITGA9, the receptor of VCAM1. Finally, among the top downregulated DEGs were two genes coding for matrix-metalloproteases (MMPs), MMP1 and MMP3, which are involved in the remodeling of the extracellular matrix (ECM). The list of significant DEGs and associated literature from infected *versus* noninfected animal-derived LR-MSCs is provided in S1 Table. To confirm the interpretation of our findings, we used a computational method for pathway enrichment analysis of the transcriptional profiles of LR-MSCs over RSV infection. Gene set enrichment analysis (GSEA) revealed that the transcriptional signature of LR-MSCs isolated from RSV- and mock-infected animals is retained during the short *ex vivo* culture phase and is following the different phases of respiratory disease. During the acute phase of

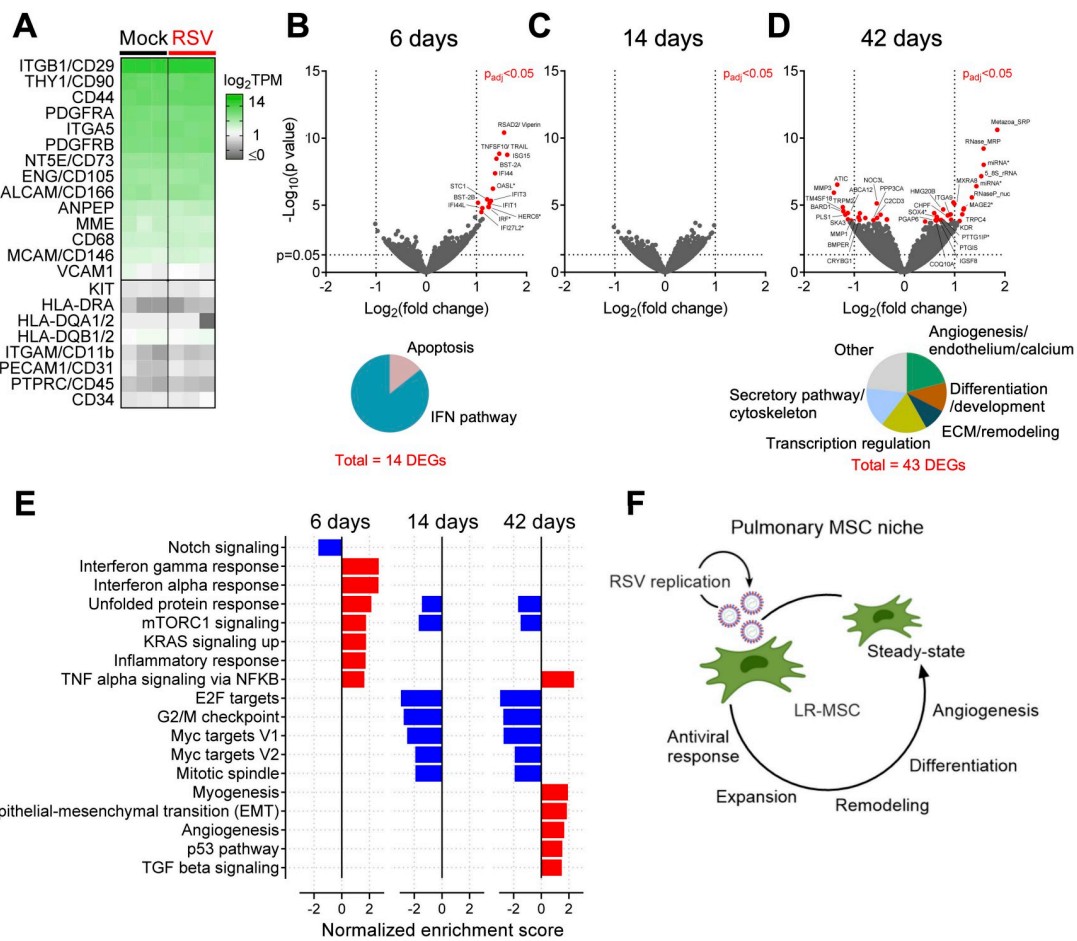

**Fig 6. The transcriptional activity of LR-MSCs is related to the phases of RSV disease.** (A) Expression levels of selected classical MSC markers expressed as $\log_2$TPM (TPM, transcripts per million). Each column is a different time p.i. (6, 14, and 42 days p.i.). (B-D) Volcano plots of the DEGs from the comparison of RSV infection and mock control 6 (B), 14 (C), and 42 (D) days p.i. with $p_{adj}$<0.05 (red filed circles). (B) Pie chart with the 14 DEGs 6 days p.i. ($p_{adj}$<0.05) categorized according to their function. (D) Pie chart with the 43 DEGs 42 days p.i. ($p_{adj}$<0.05) categorized according to their function. (E) Normalized enrichment score of significant (false discovery rate, FDR<0.05) hallmark gene sets for the comparisons of RSV infection and mock control 6, 14, and 42 days p.i. (downregulated, blue and upregulated, red) (F) Illustration summarizing the different functions of LR-MSCs over the course of RSV disease.

infection, the significant enriched hallmark gene sets were associated to IFN pathway, inflammation, and stress response. Notably, the only enriched gene set downregulated in the acute phase of RSV infection was Notch signaling, a central pathway regulating cell proliferation and differentiation. During the recovery and the convalescence phases, shared gene sets associated with cell cycle regulation, such as E2F and Myc pathway, were downregulated compared to mock controls. Further, during the convalescence phase of RSV disease, five additional hallmark gene sets were upregulated. More precisely, the transcriptional signature of LR-MSCs isolated from infected animals long after viral clearance revealed processes such as myogenesis, epithelial-mesenchymal transition (EMT), angiogenesis, p53 pathway, and TGF-β signaling, suggesting that LR-MSCs are involved in tissue repair and regeneration following virus-induced injury (Fig 6E). To sum up, these results show that LR-MSCs are activated during virus-induced lung injury by an increase in their proliferative activity and by mounting dynamic transcriptional profiles leading to an early antiviral and inflammatory response followed by mechanisms associated with tissue remodeling, repair and regeneration (Fig 6F).

## Discussion

This work was undertaken to explore the role of the pulmonary MSC compartment during acute respiratory virus infection and its contribution to lung repair. Our investigations herein reveal upon infection, LR-MSCs mount an antiviral response and release a variety of immuno-modulatory mediators, which may have a biological impact on the pulmonary microenvironment. Furthermore, we show that RSV-mediated lung injury activates and stimulates the expansion of LR-MSCs which mount a dynamic transcriptional program related to mechanisms of repair and regeneration.

Contrary to the well-described replication of RSV in AECs, the primary cellular target of RSV infection [21–23], our data indicates a distinct life cycle in LR-MSCs with viral spread via cell-to-cell contact. Such mechanism may improve virus dissemination and promote immune evasion [41]. In a humanized mouse model, pulmonary mesenchymal cells were shown to be susceptible to different viruses such as Middle East respiratory syndrome coronavirus, Zika virus, and cytomegalovirus [42]. In addition, they were recently proposed as an immune-privileged niche for *M. tuberculosis* persistence [43]. Since LR-MSCs are described to localize perivascular and in close proximity to the respiratory and alveolar epitheliums, this makes them a potential nonepithelial target for virus infection. Here, we show accessibility of LR-MSCs possibly through physical disruption of the alveolar epithelium. Indeed, already 3 days p.i., we observed evidence of lung injury associated with alveolar infection and concomitant LR-MSCs targeting by the virus. RSV infections can lead to a number of extrapulmonary manifestations [44]. Assuming the presence of MSCs in most tissues, it is conceivable that in some circumstances, RSV may target other MSC niches throughout the body. In line with this, bone marrow and umbilical cord vein MSCs were shown to be susceptible to RSV infection *in vitro* [45,46].

During the acute phase of RSV disease, we observed RSV-positive multinucleated LR-MSCs in the BAL-expanded cultures, indicative of a cytopathic effect on these cells. Thereby, the concomitant expansion of the pulmonary MSC compartment is potentially indicative of a replenishment of the lost fraction of LR-MSCs following RSV infection. Furthermore, we show that RSV infection is signaled by LR-MSCs through the induction of antiviral and pro-inflammatory mediators, both associated with RSV disease severity [47–49]. In addition, RSV infection of LR-MSCs led to a type I and III IFNs release and ISGs induction, shifting LR-MSCs towards an antiviral state. This is in line with the pro-inflammatory and antiviral transcriptional signatures found in LR-MSCs, isolated from infected animals during the acute phase of RSV disease. In mice, it has been shown that IFN signaling of the lung stromal compartment was protective against influenza virus by reducing viral replication [50]. Massive lung infiltration of immune cells following cytokine release, termed "cytokine storm", is characteristic of severe respiratory virus infections including RSV [51]. Among the most abundant chemokines in infants suffering from RSV bronchiolitis are CXCL10/IP-10, CXCL8/IL-8, CCL2/MCP-1 and CCL3/MIP-1α. Similarly, in our setting, RSV infection of LR-MSCs led to a massive release of chemokines with the potential to attract neutrophils, eosinophils, monocytes, and lymphocytes, such as CXCL8/IL-8, CCL5/RANTES, CCL2/MCP-1, and CXCL10/IP-10, respectively. In RSV disease, it is believed that the cellular sources of the pulmonary chemokine responses are epithelial and inflammatory immune cells; our data suggests that LR-MSCs might provide an additional source [52].

Besides the vast chemokine production, RSV infection of LR-MSCs led as well to the release of a number of other mediators and growth factors. Interestingly, the secretory profile of RSV-infected LR-MSCs was very distinct to the one described for RSV infected BM-MSCs, supporting tissue specificity of MSC populations [45,53,54]. Notably, VEGF, a key mediator of

angiogenesis, is released by pediatric LR-MSCs at increased levels after RSV infection. Also, the transcriptional profiles of LR-MSCs isolated during the convalescence phase of RSV disease were related to endothelial biology and pathway analysis revealed pro-angiogenic enriched gene sets. In line with this, using an adult rat model, VEGF signaling disruption was shown to cause an enlargement of the air spaces and an alteration of the alveolar structure, suggesting that VEGF has a crucial role during normal adult alveolarization [55]. Considering the perivascular location of MSCs in tissues, our data indicate that LR-MSCs may modulate endothelial cell activity during virus-induced lung injury.

TGF-β is a central regulator of the respiratory system and controls epithelial and mesenchymal cell fate through several mechanisms such as promotion of EMT, myofibroblast differentiation, and stimulation of ECM production and remodeling [56]. Our transcriptional data from LR-MSCs isolated from infected animals during the convalescence phase of RSV disease, revealed that these cells might play a role in these mechanisms. Indeed, compared to noninfected control-derived LR-MSCs, their transcriptome profiles are linked with TGF-β signaling and EMT. The latter is linked with the acquisition of more proliferative stem-like states providing an additional link between activation of EMT and wound healing [57]. In line with this, the p53 pathway activation further supports the maintenance of MSC stemness for subsequent repair processes [58]. Transcriptome analysis of LR-MSCs isolated long after virus clearance, positively associated with myogenesis. Myofibroblasts, an activated type of mesenchymal cells and an important source of ECM, participates in tissue repair but their persistence after wound healing is linked to fibrotic disorders [59]. Thus, failure of these mechanisms may potentially increase the risk of developing fibrotic abnormalities following respiratory virus infections. Comorbidities such as recurrent wheezing and asthma are happening years after RSV infection has resolved. To date, it remains unclear if pulmonary mesenchymal cells deficiency in lung repair and regeneration are linked with such disorders [60,61].

Due to their immunomodulatory properties, MSCs are applied in cell-based therapies with promising outcomes for the treatment of pulmonary morbidities such as idiopathic pulmonary fibrosis (IPF), acute-respiratory distress syndrome (ARDS) and severe influenza infections [14,62–65]. Furthermore, the current coronavirus disease 2019 (COVID-19) pandemic is leading to a strong interest in MSCs as a treatment option for severe COVID-19 cases. As a consequence, an increasing number of clinical investigations of such MSC-based therapy approaches are under evaluation [66–69]. Safety concerns regarding susceptibility of MSCs towards different viruses, as well as potential viral reactivation events have been raised [70]. On one hand, our data indicate potential safety concerns due to the targeting of LR-MSCs during the acute phase of virus infection. On the other hand, we provide evidence that LR-MSC seem to have a beneficial effect after virus clearance, supporting the use of MSC-based therapies to treat virus-induced lung injury.

The ovine lung is a classical model of the human respiratory tract due to similarities in size, structure, development and immune system [71]. However, some limitations should be mentioned. First, we observed a high animal-to-animal variability in most endpoints measured, probably due to the outbred nature of the animals. While this might look like a limitation, we believe it reflects best the situation in humans compared to inbred models. Secondly, we used the isolated LR-MSCs as a single population but previous single-cell transcriptomic studies demonstrated that MSC, are a rather heterogeneous population composed of distinct MSC subsets [8,9,72]. Third, due to the lung size, we analyzed three pooled specimens, representative of each lung region. This might explain why we didn't detect RSV-positive cells for some animals for whom infection was demonstrated by the assessment of the viral loads.

In summary, our data demonstrate the involvement of pulmonary MSCs during respiratory virus infection. While being a target for RSV, these cells can respond to infection by switching

to an antiviral state and by a profound remodeling of their immune phenotype. Furthermore, LR-MSCs show hallmarks of tissue repair and regeneration processes after viral clearance. Our findings identify a function of LR-MSCs in a highly prevalent clinical situation and constitute a basis for further exploration of the pulmonary mesenchymal compartment during virus infections.

# Material and methods

## Ethics statement

The experiments with lambs were performed in compliance with the Swiss animal protection law (TSchV SR 455.1; TVV SR 455.163). They were reviewed by the cantonal committee on animal experiments of the canton of Bern, Switzerland and approved by the cantonal veterinary authority with the license number BE125/17 (Amt für Landwirtschaft und Natur LANAT, Veterinärdienst VeD, Bern, Switzerland). For the isolation of the RSV-ON1-H1 strain, the caregivers gave informed consent for the donation of nasopharyngeal aspirates, and all steps were conducted in compliance with good clinical and ethical practice and approved by the local ethical committee at Hannover Medical School, Germany (permission number 63090/2012). The isolate was passaged up to five times in HEp-2 cells (ATCC, CCL-23). For the isolation of primary pediatric MSCs, written informed consent was obtained for all the patients and/or parent/guardian and the study protocol was approved by the local Ethics Commission of the Canton of Bern, Switzerland (KEK-BE:042/2015).

## Virus propagation and titration

Human RSV-A2 strain was derived from ATCC (VR-1540, GenBank accession number KT992094.1). The recombinant RSV-mCherry is described elsewhere [73] and the RSV-GFP construct was generated by Mark Peeples (Nationwide Children's Hospital Columbus, USA) [74] and kindly provided by Dominique Garcin (University of Geneva, Switzerland). The clinical RSV isolate, RSV-ON1-H1, was isolated from a child below the age of five years with confirmed RSV infection, hospitalized at Hannover Medical School, Germany [75]. All RSV strains were propagated on HEp-2 cells, cultivated in DMEM (Gibco) supplemented with 10% fetal bovine serum (FBS, Gibco). Briefly, HEp-2 cells were infected at a MOI of 0.02 PFU/cell for 2 hours before the addition of DMEM supplemented with 5% FBS. The cell-associated virus was harvested by scraping the cell monolayer when 60% of cytopathic effect was observed. The virus was released from the cells by one freeze-thaw cycle followed by centrifugation to get rid of cell debris. Virus stocks were stored in aliquots containing 10% sucrose (Sigma) at -150˚C. To prepare a mock control the same procedure was applied without adding virus. A similar approach was applied to assess the intracellular RSV titers from infected LR-MSCs. For virus titration, serial dilutions of supernatants were added to HEp-2 cells and incubated for 48–96 h at 37˚C, 5% $CO_2$. Next, the cells were washed with PBS, fixed with methanol supplemented with 2% $H_2O_2$ and incubated with a biotinylated anti-RSV antibody (Bio-Rad) in PBS containing 1% bovine serum albumin (BSA, Sigma) for 1 h, followed by 30 min incubation with ExtrAvidin Peroxidase (Sigma) and staining with 3,3′-diaminobenzidine substrate (Sigma). Virus titers were expressed as PFU/ml.

## Primary cells

Primary pediatric MSCs were obtained as described previously [32,72]. Briefly, healthy lung tissue was obtained from pediatric patients undergoing elective surgery for congenital pulmonary airway malformation and other various airway abnormalities (n = 9, age range: 5 days to

181 months; S2 Table). Minced tissue was further disaggregated enzymatically using a solution containing collagenase I and II (Worthington Biochemicals) at a concentration of 0.1 and 0.25%, respectively. Further, digestion buffer was supplemented with 0.2 mg/ml deoxyribonuclease I (Biochemicals) to the digestion buffer to prevent cell clumps and improve cell recovery. To isolate MSCs, a FCM assay was applied using a panel of fluorescently-conjugated human monoclonal antibodies: CD45, CD14, CD31, and EpCAM for negative and CD73 and CD90 for positive selection. FCM-sorted MSCs were expanded in tissue culture flasks precoated with 0.2% gelatin solution (Sigma) and in chemically defined growth medium consisting of α-MEM with ribonucleosides (Sigma) supplemented with 1% FBS, GlutaMAX (Invitrogen), 10 ng/ml of recombinant human fibroblast growth factor 2 (FGF2, Life Technologies), 20 ng/ml of recombinant human epidermal growth factor (EGF, Life Technologies), human insulin (Sigma), 100 units/ml of penicillin and 100 μg/ml streptomycin (Sigma). Cells were maintained in a humidified atmosphere at 37°C, 5% $CO_2$ until confluence was reached. The medium was aspirated carefully and replaced with fresh medium six days after plating and after that point performed twice a week. Cells were used prior to passage 5 in all experiments. WD-AECs isolated from independent healthy donors (MucilAir, Epithelix Sàrl, Geneva, Switzerland) were used for RSV infection. The cultures were maintained on 24-well transwell inserts (Corning) at the air-liquid interface (ALI) in MucilAir culture medium (Epithelix Sàrl, Geneva, Switzerland) in a humidified incubator at 37° C, 5% $CO_2$. Every 3 days culture medium was changed. The apical surface was rinsed with Hank's balanced salt solution (HBSS, ThermoFisher) once a week and prior to infection, to washout the mucus.

## Precision-cut lung slices

Isolated ovine lungs were infused with 1.5% low-melting point agarose (Promega) in DMEM and subsequently put into cold PBS in order to allow the agarose to solidify. Next, the perfused lung tissue was cut into pieces of 0.1–1 $cm^3$ and embedded in 4% low-melting point agarose. An automated vibrating microtome (VT1200S, Leica Biosystems) was used at a speed of 0.1 mm/sec and an amplitude of 2.5–2.8 mm to generate PCLS with a thickness of 200 μm. Uniformly sized slices were punched at a diameter of 8 mm. The PCLS were maintained in DMEM, supplemented with 1% FBS, 100 units/ml of penicillin and 100 μg/ml streptomycin, and 2.5 μg/ml of Amphotericin B (all from ThermoFisher) in a humidified atmosphere at 37°C, 5% $CO_2$. The culture medium was changed every 24 h and the PCLS cultures were infected 2–3 days following preparation.

## RSV infection

To perform infection of primary pediatric LR-MSCs and AECs with RSV-A2, RSV-mCherry, or RSV-ON1-H1, $10^5$ cells per well of a 12 well plate were seeded 24 h prior to infection and/or stimulation. Cells were infected at selected MOIs of RSV or mock-control in FBS-free medium for 1 h at 37°C and 5% $CO_2$. Subsequently, cells were washed three times with 1X PBS and then incubated in growth medium. Ten μg/ml of poly(I:C) (Sigma) was applied in growth medium and incubated alike until harvesting. At selected time points, supernatants and cells were harvested and stored appropriately for further analysis. For infection of WD-AECs, virus preparations were diluted in medium and virus dilutions were applied apically to the tissue, assuming an average of 500'000 cells per insert. As a mock control, the culture supernatant of mock-infected HEp-2 cells was used. Virus particles were allowed to adsorb for 3 h at 37°C and 5% $CO_2$ before the inoculum was removed and each insert was carefully washed three times with pre-warmed HBSS and placed on a new well containing fresh MucilAir culture medium. For stimulation of the WD-AEC inserts, 10 μg/ml poly(I:C) were applied to the

basolateral chamber containing the culture medium. For examination of infectious viral particle release 2, 24, 48, 72, and 144 hours p.i., 250 μl HBSS was added to the apical surface and incubated for 20 min at 37˚C, harvested, and stored at −70˚C until further analysis. Basolateral media was collected and similarly stored, and replaced with fresh media in the basolateral chamber of the inserts. For the infection of ovine PCLS, $5x10^5$ PFU of RSV-GFP per PCLS were applied in growth medium, supplemented with 0.1% FBS and incubated for 24 h at 37˚C and 5% $CO_2$. Subsequently, the infection medium was changed to growth medium and media was changed every 24 h.

## RNA isolation and quantitative PCR

Total RNA of primary pediatric LR-MSCs, human AECs, and ovine lung- and BAL-derived cultured MSCs was extracted using the Nucleospin RNA Plus Kit (Macherey-Nagel) according to the manufacturer's protocol. For the isolation of the total RNA from the supernatants of infected LR-MSCs, we used the QIAamp Viral RNA kit following manufactures recommendations (Qiagen). For reverse transcription and synthesis of complementary DNA the Omniscript RT Kit (Qiagen) using random hexamers (Invitrogen) was applied. Quantitative PCR was performed with target-specific primers using the TaqMan Gene Expression Assay (Applied Biosystems) or the Fast SYBR Green Assay (ThermoFisher) on an ABI Fast 7500 Sequence Detection System (Applied Biosystems). For lung tissue and the BAL cellular fraction quantification was performed using AgPath-ID One-Step RT-PCR Reagents (ThermoFisher), according to the manufacturer's instructions. The data were analyzed using the SDS software (Applied Biosystems). Relative expression was calculated with the ΔΔCT method [76]. The expression levels of the genes of interest, were normalized to the housekeeping 18S rRNA. For the analysis of viral loads, RNA copy numbers were interpolated from a standard curve generated with the serial dilution of a plasmid containing the cDNA of the RSV L gene or the housekeeping 18S rRNA. The sequence of all the primers and probes is summarized in S3 Table.

## Flow cytometry

FCM assays using different multicolor staining protocols were applied for the analysis of co-stimulatory molecule expression and RSV infection of primary pediatric LR-MSCs, for the identification and analysis of RSV infection of different cell types in lung cell suspensions derived from lambs, the assessment of apoptosis in the lung and the alveolar space of infected and mock-infected animals. Further, a combination staining was applied to evaluate the phenotype of cultured LR-MSCs. The detailed list of reagents used is summarized in S4 Table. FCM acquisitions were performed using a BD FACS Canto II (BD Bioscience) using the DIVA software and were further analysed with the FlowJo software version 10.6.0 (BD). Dead cells were excluded by electronic gating in forward/side scatter plots, followed by exclusion of doublets as well as LIVE/DEAD Fixable Dead Cell Staining (Invitrogen). Permeabilization and fixation were undertaken according to the manufacturer's protocol (BD and Invitrogen).

## Trans-differentiation assay

For differentiation, pediatric LR-MSCs (passage 5) and ovine LR-MSCs (passage 3) were plated in a 24-well plate in regular culture medium and placed in a humidified atmosphere at 37˚C with 5% $CO_2$. After 48 hours, cells were washed with PBS and complete differentiation media of the different StemPro Differentiation Kits (Thermofisher) was added. For adipogenic induction cells were plated at $10^4$ cells/cm$^2$, for chondrogenic induction cells were plated at $2.7x10^4$ cells/cm$^2$, and to induce osteogenic differentiation, cells were plated at $5x10^3$ cells/cm$^2$.

Medium was changed twice a week and after 15–21 days, cells were fixed with 4% PFA (Sigma). Adipocytes were stained with Oil Red O (Sigma) to detect the formation of lipid droplets and counterstained with hematoxylin (Carl Roth). Chondrocytes were stained with Alcian Blue (Sigma) to visualize glycosaminoglycan synthesis and Alizarin Red S staining (Sigma) was used to detect calcium deposits in osteocytes.

## Microscopy analysis

Ovine BAL-derived MSCs and expanded pediatric LR-MSCs were cultured on a Nunc Lab-Tek II Chamber Slide System with 50'000 cells per chamber (ThermoFisher). Next, the cells were washed with cold PBS with $Ca^{2+}$ and incubated with Wheat Germ Agglutinin (WGA)-AF633 (ThermoFisher) for 10 min at 4°C. Then, cells were fixed with 4% PFA for 10 min at room temperature. Next, the cells were washed with 0.3% saponin (Sigma) and stained with RSV-fusion protein antibody (Millipore) for 20 min at 4°C. Then, 20 min incubation at 4°C with anti-mouse IgG2a-AF546 or -AF488 (both ThermoFisher) was applied. Finally, DAPI (Sigma) was added for 5 min at 37°C and cells were washed three times with cold PBS with $Ca^{2+}$ before mounting with MOWIOL 4–88 Reagent (Sigma). The detailed list of antibodies used is summarized in S4 Table. Confocal microscopy analysis was performed with a Nikon Eclipse Ti microscope (Nikon). All images were acquired using a 63X oil-immersion or a 40X objective. The images were analyzed with IMARIS 7.7 software (Bitplane) with threshold subtraction and gamma correction. To monitor the spread of RSV-mCherry infection in WD-AEC in comparison to MSC cultures, a Nikon BioStation CT was used (Nikon). The cultures were followed over a period of 2 days with a 10X objective and images were acquired in an automated sequence every 4 to 8 hours following infection. Reporter expression of RSV-mCherry-infected pediatric LR-MSCs, ovine LR-MSCs, and RSV-GFP-infected ovine PCLS was assessed using an Evos FL Auto 2 cell imaging system (ThermoFisher) or a Leica TCS-SL. The micrographs from the trans-differentiation assays were captured using an inverted microscope ECLIPSE TS100 with a Nikon DS-Fi3 camera using a DS-L4 application v.1.5.03 (all from Nikon).

## Immunoassays

When performing cytokine quantification, the supernatants and the basolateral medium from RSV-A2 and RSV-ON1-H1 infections as well as of poly(I:C)-stimulated and unstimulated, and mock-infected pediatric LR-MSCs and WD-AECs, respectively, were harvested after 24 and 72 h of culture at 37°C with 5% $CO_2$. Human IFN-β and IFN-λ1/3 concentrations were determined using commercial enzyme-linked immunosorbent assay (ELISA) kits (R&D Systems) following the manufacturer's protocol. For the multiplex assay, the Pro Human Cytokine 27-plex Assay (Bio-Rad) was used according to the manufacturer's protocol and read on a Bio-Plex 3D suspension array system including a Bio-Plex Manager software v 6.0 (Bio-Rad).

## Animals

Newborn lambs, males and females were allocated randomly to different groups (see below). Before and during the experiment, they were kept with their ewes. In order to prevent secondary bacterial infections, the lambs received a prophylactic long-acting antibiotic treatment by intramuscular injection of an oxytetracycline hydrochloride preparation (Cyclosol LA, 20 mg/kg) starting at 18 to 24 h before RSV infection. The treatment was repeated 2 times with 4 days interval. The trans-tracheal RSV inoculation was performed under sedation and analgesia by intramuscular injection of a mixture of midazolam (Dormicum) 0.2 mg/kg and butorphanol (Morphasol) 0.2 mg/kg. The trachea was punctured between the tracheal rings with a 0.9 x 40

mm needle (20G) and a volume of 2 ml of virus ($10^8$ PFU) or PBS (mock-infected controls) were injected. Body temperature, body weight and clinical status (respiratory and heart rates) were assessed daily by a veterinarian, whenever possible by the same person to ensure unbiased clinical assessment. At defined time points, namely at 3, 6, 14, and 42 days p.i., the lambs were euthanized and examined for gross- and histopathology. The lung was processed for *post-mortem* bronchoalveolar lavage.

## Histopathological evaluation of lung tissue

Lung specimens were fixed in 4% buffered formalin for 48 h, embedded in paraffin and routinely processed for histology. Histological sections of 3 μm were stained with hematoxylin and eosin coloration and observed by light microscopy. For immunohistochemistry, deparaffinization of the sections was done with xylol for 5 minutes followed by rehydration in descending concentrations of ethanol (100, 95, 80, and 75%). $H_2O_2$ (3.25% in methanol, 10 min at room temperature) inhibited endogenous peroxidase activity. Then, the slides were incubated in boiling citrate buffer (pH 6.0) for 10 min for antigen retrieval. 1% BSA (10 min) was used for blocking of nonspecific antibody binding, followed by an overnight incubation at 4˚C with the primary antibody targeting RSV (ThermoFisher). For secondary antibody incubation and signal detection LSAB and AEC Kits (DakoCytomation) were used following the manufacturers protocol. Counterstaining was done with Ehrlich hematoxylin and cover slips were mounted using Aquatex (Merck) [77].

## Processing of lung tissue and bronchoalveolar lavage

One specimen per lung region (cranial, middle, and caudal) were pooled and dissociation was done using a collagenase I and II and a DNase I enzyme mix (all from BioConcept) and the gentleMACS Octo Dissociator (Miltenyi Biotec). Following this mechanical and enzymatical dissociation, the samples were applied to a sieve, to remove any remaining particulate matter. The cell suspensions were passed through cell strainers (100 and 70 μm pore-size, Falcon) and centrifuged at 350g for 10 min at 4˚C to obtain single-cell suspensions. For the isolation of cells from BALs, the lungs were isolated with the trachea, which was clamped before cutting, to prevent blood from entering the lungs. Then, a PBS-containing antibiotic solution with 100 units/ml of penicillin and 100 μg/ml streptomycin (both Sigma), and 2.5 μg/ml Amphotericin B was poured into the lungs through a sterile funnel (200–500 ml). The cell suspensions were then passed through cell strainers (100 and 70 μm pore-size, Falcon) and centrifuged at 350g for 10 min at 4˚C to obtain single cell suspensions. If needed, red blood cells were lysed by resuspending the pellet with 2 ml of $H_2O$ and washed immediately in cold PBS before centrifugation at 350g for 10 min at 4˚C. Cells were then processed for FCM analysis, CFU-F assay or expanded in culture to obtain lung-derived MSCs. To isolate and expand MSCs in culture, lung cell suspensions were seeded at a density of $3.5x10^4$ cells per $cm^2$ in tissue culture flasks as described previously [3]. Cells were maintained in α-MEM (Thermofisher) supplemented with 10% FBS (ThermoFisher), an antibiotic solution containing 100 units/ml of penicillin and 100 μg/ml streptomycin, and 2.5 μg/ml Amphotericin B. After one to two days, the medium was changed and cells were maintained at 37˚C with 5% $CO_2$ until reaching ~80% of confluence. Regular media changes were performed twice a week.

## Colony-forming unit-fibroblast assay

Lung single-cell suspensions were seeded at a density of $3.5x10^4$ cells per $cm^2$ in a six-well plate as described previously [3]. Cells were fixed after 7–14 days with ice-cold methanol for 20 min and then washed with PBS. The cells were stained with Giemsa stain for 6 min, rinsed with

H₂O, and air-dried. Images were captured using an ImmunoSpot analyzer (CTL). Two experienced investigators performed the CFU-F counts independently.

## RNA isolation and mRNA sequencing and data analysis

For mRNA sequencing, total RNA was extracted from ovine LR-MSCs using TRIzol reagent (ThermoFisher) in combination with the Nucleospin RNA Kit (Machery-Nagel) as previously described [78]. In short, cells were lysed with 1 ml of TRIzol reagent and kept at -70˚C until further processing. After thawing, 0.2 ml chloroform was added to the TRIzol lysate and the samples were mixed vigorously and incubated for 2–3 min at room temperature. The extractions were then centrifuged at 12'000g for 15 min at 4˚C. The aqueous phase was collected and mixed with 500 μl 75% ethanol and the RNA precipitated for 10 min at room temperature. The RNA precipitate was further purified with the Nucleospin RNA kit according to the manufacturer's instructions. The quantity and quality of the extracted RNA was assessed using a ThermoFisher Scientific Qubit 4.0 fluorometer with the Qubit RNA BR Assay Kit (Thermo Fisher Scientific) and an Advanced Analytical Fragment Analyzer System using a Fragment Analyzer RNA Kit (Agilent), respectively. Thereafter, cDNA libraries were generated using an illumina TruSeq Stranded mRNA Library Prep (illumina) in combination with IDT for Illumina–TruSeq RNA UD Indexes (Illumina). The illumina protocol was followed using the recommended input and quality of total RNA. The quantity and quality of the generated NGS libraries were evaluated using a Thermo Fisher Scientific Qubit 4.0 fluorometer with the Qubit dsDNA HS Assay Kit (ThermoFisher) and an Advanced Analytical Fragment Analyzer System using a Fragment Analyzer NGS Fragment Kit (Agilent), respectively. The library pool was paired end sequenced using a NovaSeq 6000 SP Reagent Kit v1.0, 100 cycles (illumina) on an Illumina NovaSeq 6000 instrument. The quality of the sequencing runs was assessed using illumina Sequencing Analysis Viewer (illumina version 2.4.7) and all base call files were demultiplexed and converted into FASTQ files using illumina bcl2fastq conversion software v2.20. The average number of reads per library was 33.5 million. The RNA quality-control assessments, generation of libraries and sequencing runs were performed at the Next Generation Sequencing Platform, University of Bern, Switzerland. Analysis of the RNA-seq data was performed at the Interfaculty Bioinformatics Unit at the University of Bern, Switzerland. RNAseq data quality was assessed using fastqc v. 0.11.5 and RSeQC v. 2.6.4 [79]. The reads were mapped to the *Ovis aries* reference genome (Oar_v3.1) with Hisat2 v.2.1.0 [80]. To count the number of reads overlapping with each gene, as specified in the Ensembl annotation FeatureCounts from Subread v.1.5.3 [81] was used. The Bioconductor package DESeq2 [82] was applied to test for differential gene expression between the experimental groups. To evaluate gene expression levels of MSC markers we applied the TPM normalization method [83]. The molecular signatures database (MSigDB) with the Hallmark gene sets was used for GSEA [84]. Analysis was done using the Prism 8 software (GraphPad).

## Statistical analysis

The Prism 8 software was used for statistical analysis. To determine differences between two groups, non-parametric paired Wilcoxon test or Mann–Whitney U-test were used. For multiple comparisons one-way ANOVA with Tukey post-hoc test or Kruskal-Wallis test with Dunn's correction were used. A $p < 0.05$ was considered statistically significant.

## Supporting information

**S1 Fig. Infection of pediatric LR-MSCs with RSV at a MOI of 1 PFU/cell.** (A) RSV F-protein positive LR-MSCs assessed by FCM and plotted over time. LR-MSCs were infected with 1

PFU/cell with RSV-A2 (n = 3) or a clinical isolate RSV-ON1-H1 (n = 4–6). Each symbol represents an individual donor. (B) Supernatants of infected LR-MSCs or apical washes of infected WD-AEC cultures were analyzed by a PFU assay. Cells were infected with RSV-ON1-H1 at a MOI of 1 PFU/cell. A Mann-Whitney U test was applied to compare the two cell types (WD-AECs, n = 3 *versus* LR-MSCs, n = 5–6). Each symbol represents an individual donor. *p<0.05, **p<0.01. (C) Intracellular infectious RSV titers over time in LR-MSCs infected with RSV-A2 at 1 PFU/cell. Each symbol represents an individual donor (n = 3). (D) Extracellular viral RNA loads over time in supernatants of infected LR-MSCs or apical washes of infected WD-AEC cultures. Cells were infected with RSV-ON1-H1 at a MOI of 1 PFU/cell. A Mann-Whitney U test was applied to compare the two cell types (WD-AECs, n = 3 *versus* LR-MSCs, n = 4–6). Each symbol represents an individual donor. *p<0.05.
(TIFF)

**S2 Fig. IFN mRNA levels and PD-L1 and MHC class I surface expression in RSV-infected pediatric LR-MSCs.** (A-C) mRNA levels of IFN-β (A), IFN-λ1 (B), and IFN-λ2/3 (C) in LR-MSCs and WD-AECs infected with mock control or RSV-A2 at a MOI of 1 PFU/cell for 24 and 72 hours. Boxplots indicate the median value (centerline) and interquartile ranges (box edges), with whiskers extending to the lowest and the highest values. Each symbol represents an individual donor (LR-MSCs, n = 4–6; WD-AECs, n = 3). The data were compared with the Kruskal–Wallis test followed by Dunn's post hoc test. (D) Representative histogram of the surface expression of PD-L1 and MHC class I in pediatric LR-MSCs 24 h post-treatment with mock, poly(I:C) 10 µg/ml, and RSV-A2 at 1 PFU/cell. (E) Median fluorescence intensity (MFI) of PD-L1 and MHC class I expression. Each symbol represents an individual donor (n = 3). The data were compared with the Kruskal–Wallis test followed by Dunn's post hoc test. *p<0.05.
(TIFF)

**S3 Fig. RSV-infection of ovine PCLS and ovine LR-MSCs.** (A) Representative histograms showing expression of the surface markers CD29, CD44, and CD166. (B) Representative phase-contrast (PC) micrograph showing morphology in culture and demonstrates plastic adherence. Representative images of Toluidine blue, Alizarin Red S, and Oil Red O stainings after chondrogenic (C), osteogenic (O), and adipogenic (A) differentiation, respectively. Magnification 100X (PC, O), 200X (C, A). (C) Cell-associated RSV RNA loads expressed as RSV copies per $10^9$ 18S copies (red empty circles) and infectious virus release in PFU per ml (blue empty circles) over time following infection of primary ovine LR-MSCs with 0.1 PFU/cell of RSV-A2 determined 24, 48, and 72 h p.i. Each symbol represents an individual donor (n = 3). The positive control (P) is the virus preparation used for the infections having a titer of $2x10^7$ PFU/ml. M, mock. (D-E) Ovine PCLS infected with RSV-GFP at $5x10^5$ PFU per PCLS. Representative fluorescence micrographs are shown at 72, 120, and 144 hours p.i. Scale bar, 650 µm (D). Representative fluorescence micrographs indicating infection of pneumocytes. Scale bar, 125 µm (E).
(TIFF)

**S4 Fig. Presence of potential syncytium in neonatal lungs following RSV infection.** (A, B) Representative histopathological sections of the lung tissue from lambs 6 days p.i. infected with RSV-A2. The red and blue arrowheads indicate the presence of potential syncytia and the green arrowheads indicate dome-shaped type 2 alveolar cells lining the alveolar wall, indicative for type 2 alveolar cell hyperplasia. Scale bar, 50 µm.
(TIFF)

**S5 Fig. FCM assay for apoptosis detection *in vivo*.** Gating strategy for the detection of cleaved caspase-3-positive cells. G1, gate 1.
(TIFF)

**S6 Fig. Infection of epithelial cells and LR-MSCs following neonatal RSV infection.** RSV-positive epithelial cells (Epith) and LR-MSCs in lung cell suspensions were detected with an FCM assay 3, 6, 14, and 42 days following RSV-A2 infection of neonates. The dashed line depicts the detection limits (0.6% for Epith and 0.8% for LR-MSCs). Boxplots indicate the median value (centerline) and interquartile ranges (box edges), with whiskers extending to the lowest and the highest values. Each symbol represents an individual animal (mock, n = 3–4; RSV, n = 5–8).
(TIFF)

**S7 Fig. Formation of syncytium upon RSV infection of human and ovine LR-MSCs.** (A, B) Representative micrographs of pediatric (A) and ovine (B) LR-MSCs infected with 0.1 PFU/cell of RSV-mCherry 48–72 hours p.i. Giant multinucleated cells, indicative of syncytium formation, were observed. Yellow arrowheads indicate clusters of nuclei and the yellow dashed line indicates a giant multinucleated cell. PC, phase-contrast. Magnification 100X (PC, pediatric LR-MSCs) and 400X (PC and RSV, ovine LR-MSCs). Scale bar, 125 μm.
(TIFF)

**S1 Table. List of significant DEGs in RSV-infected *versus* mock-treated animals.**
(DOCX)

**S2 Table. Patient characteristics.**
(DOCX)

**S3 Table. List of qPCR primers.**
(DOCX)

**S4 Table. List of antibodies used.**
(DOCX)

## Acknowledgments

We are grateful to Sylvie Python and Aurélie Godel for technical assistance; Inês Margarida Berenguer Veiga, Nathan Leborgne, Loic Borcard, Amal Fahmi and Charaf Benarafa for participation in the animal trial; Hans-Peter Lüthi, Roman Troxler, Jan Salchli, Daniel Brechbühl, Katarzyna Sliz, and Veronika Ayala for animal care. We thank Dominic Garcin for sharing the RSV-mCherry construct and Mark E. Peeples for the RSV-GFP construct. We thank Sibylle Haid, Martin Wetzke, Gesine Hansen, and Thomas Pietschmann for sharing the RSV-ON1-H1 isolate. We thank Heidi Tschanz-Lischer, Irene Keller, and Jenna Kelly for their bioinformatics support. We thank Christoph Aebi for helpful discussions and all study participants and their families.

## Author Contributions

**Conceptualization:** Nicolas Ruggli, Sean R. R. Hall, Marco P. Alves.

**Data curation:** Melanie Brügger, Thomas Démoulins, G. Tuba Barut, Artur Summerfield, Marco P. Alves.

**Formal analysis:** Melanie Brügger, Thomas Démoulins, G. Tuba Barut, Beatrice Zumkehr, Artur Summerfield, Marco P. Alves.

**Funding acquisition:** Marco P. Alves.

**Investigation:** Melanie Brügger, Thomas Démoulins, Beatrice Zumkehr, Blandina I. Oliveira Esteves, Kemal Mehinagic, Quentin Haas, Aline Schögler, Marie-Anne Rameix-Welti, Jean-François Eléouët, Ueli Moehrlen, Thomas M. Marti, Ralph A. Schmid, Horst Posthaus, Nicolas Ruggli, Sean R. R. Hall, Marco P. Alves.

**Methodology:** Melanie Brügger, Thomas Démoulins, Beatrice Zumkehr, Kemal Mehinagic, Aline Schögler, Marie-Anne Rameix-Welti, Jean-François Eléouët, Ueli Moehrlen, Thomas M. Marti, Ralph A. Schmid, Artur Summerfield, Horst Posthaus, Nicolas Ruggli, Sean R. R. Hall, Marco P. Alves.

**Project administration:** Marco P. Alves.

**Supervision:** Marco P. Alves.

**Writing – original draft:** Melanie Brügger.

**Writing – review & editing:** Melanie Brügger, Thomas Démoulins, G. Tuba Barut, Blandina I. Oliveira Esteves, Kemal Mehinagic, Marie-Anne Rameix-Welti, Jean-François Eléouët, Ueli Moehrlen, Thomas M. Marti, Ralph A. Schmid, Artur Summerfield, Horst Posthaus, Nicolas Ruggli, Sean R. R. Hall, Marco P. Alves.

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
