## [Decision Letter · Decision Letter 0]

26 Apr 2021

Dear Dr. Alves,

Thank you very much for submitting your manuscript "Pulmonary mesenchymal stem cells are engaged in distinct steps of host response to respiratory syncytial virus infection" for consideration at PLOS Pathogens. As with all papers reviewed by the journal, your manuscript was reviewed by members of the editorial board and by several independent reviewers. In light of the reviews (below this email), we would like to invite the resubmission of a significantly-revised version that takes into account the reviewers' comments.

The reviewers raised very fundamental issues with the virology presented in this paper. While there is interest, you must provide stronger evidence of virus replication and virus spread. The concerns raised and experiments requested by Reviewer 1 and echoed by Reviewer 3 must be addressed.

We cannot make any decision about publication until we have seen the revised manuscript and your response to the reviewers' comments. Your revised manuscript is also likely to be sent to reviewers for further evaluation.

Sincerely,

Sabra L. Klein

Associate Editor

PLOS Pathogens

Carolina Lopez

Section Editor

PLOS Pathogens

Kasturi Haldar

Editor-in-Chief

PLOS Pathogens

orcid.org/0000-0001-5065-158X

Michael Malim

Editor-in-Chief

PLOS Pathogens

orcid.org/0000-0002-7699-2064

The reviewers raised very fundamental issues with the virology presented in this paper. While there is interest, you must provide stronger evidence of virus replication and virus spread. The concerns raised and experiments requested by Reviewer 1 and echoed by Reviewer 3 must be addressed.

Reviewer's Responses to Questions

**Part I - Summary**

Reviewer #1: Brugger et al. have examined the response to respiratory syncytial virus (RSV) of lung-resident (LR) mesenchymal stem and stromal cells (MSCs) in both primary differentiated pediatric MSCs and in vivo, in the lamb model of RSV infection. MSCs are thought to play a role in the alveolar niche as regulators of homeostasis and regeneration. The transcriptional response to RSV of LR-MSCs initiates with an antiviral signature but later switches to repair mechanisms of differentiation, tissue remodeling, and angiogenesis.

The investigators state (l.45) that “[i]n the alveolar niche, LR-MSCs can interact with epithelial cells (AECs), which are the primary cellular target of most respiratory viruses [12-14].” In fact, these three references all refer to airway epithelial cells, the epithelial cells in the small airways, not the alveolar epithelial cells. RSV’s target cell is the airway epithelial ciliated cells, as is influenza virus. If RSV does infect alveolar cells, that needs to be shown. This report does not, nor does it cite another report that does. Alveolar cells are very different from airway epithelial cells. And the premise (l.36) that “[l]ung-resident (LR)-MSCs can promote alveolar cell growth, differentiation, and self-renewal” would not seem to be relevant if the cells that are damaged by RSV are the airway epithelial cells, not the alveolar epithelial cells.

As described above, if the MSC cells are not a natural target cell for RSV, they may not express the RSV receptor, that might account for the poor efficiency of infection in vitro. The authors examined the MSCs for transcripts for every suggested receptor reported for RSV on immortalized cell lines, but not the one that is considered the most likely in vivo receptor, CX3CR1. CX3CR1 has been shown to be the receptor on ciliated epithelial cells in the airway epithelium.

The investigators state (l.78) that they “detected fast replication kinetics of RSV in LR-MSCs similar to levels measured in infected AECs (Fig. 1D).” They state that “after 24 to 48 hours post-infection (p.i.) around 80% of LR-MSCs were infected by RSV-A2.” But they show no evidence for that claim. Fig. 1D It shows the number of RSV genome copies, not the number of infected cells. When they do show a picture of infected cells (Fig. 1F), only 2 cells in the field of probably 200 or more cells are infected at 36 hr and possibly 2 more by 48 hr. This infection was very inefficient, and the virus did not spread from cell to cell much at all. They do not mention what moi was used in this experiment in the text, but the Fig legend says moi of 0.1 to 0.5, which should result in the infection of 10-40% of the cells by 24 hr. That has not happened.

Instead one of the infected cells has fused with many of its neighbors. The mechanism of syncytia formation does not require virus production, it only requires that the F protein reach the cell surface in its cleaved form where it can cause fusion of that cell’s membrane with that of its neighbor. In other words, this represents one infected cell. The other major cell also looks like a syncytium, though smaller.

The authors go on to show in Fig 1G and H that no infectious virus is released from the MSCs. That would explain the results in the Fig. 1F picture in which the virus did not spread from cell to cell, except by syncytia formation. Clearly, no virus is produced from the few MSCs that were initially infected. However, the syncytia could have produced many copies of the RSV genome as were detected in Figs 1C and 1D.

Such poor infectivity and low-level virus production with fusion between the few infected cells and their neighbors is reminiscent of infection of differentiated airway cultures with RSV whose attachment protein, G, had been deleted. That virus infects ciliated cells between 1% and 0.1% as efficiently as RSV expressing G does. If these MSC cells do not express the receptor for RSV, the same type of low level of infection and low yield of virus would be expected probably, mediated by the RSV F protein instead of its G protein, accounting for the low level of infection, the syncytia formation and the lack of virus spread from cell to cell in the culture.

The investigators conclude that “Altogether, these results demonstrate that primary pediatric LR-MSCs are highly permissive to RSV infection…” I would conclude the opposite.

Fig 2 compares the gene expression of MSCs and HAEs when inoculated in vitro by RSV, at both transcripts produced (A and B) and proteins secreted (C – F), showing all kinds of responses. Again, the HAEs are not alveolar epithelial cells.

There is the question, since RSV infects the small airway epithelial ciliated cells, how would the RSV reach the MSCs. Are there MSCs beneath the ciliated airway cells? The airway epithelium has tight junctions between cells and RSV is known to be exclusively shed apically (into the lumen of the airway). It is possible though that late in infection, if more of the airway epithelial cells are killed than can be replaced, the barrier may breakdown and allow virus shed into the lumen to leak through that barrier and to contact and infect underlying cells. If these underlying cells include MSCs and the infected MSCs produce mediators as shown in Fig. 2, those mediators might act on the alveolar epithelial cells that would be nearby. Is this what the authors are envisioning?

The investigators then switch to their in vivo model of RSV infection in lambs inoculated with a very large dose of RSV, 108 pfu. They describe the shedding of RSV into the BAL and lung tissue which peaks around day 6, and the resolution of the infection over time in Fig. 3. They examined cells from the lung by FCM, which is not defined, but likely to be flow cytometry.

They found (l.158) “…at 6 days p.i. few syncytial cells lining alveoli were present.” It is not clear if that means that “a” few syncytial cells were present or that none were present. That should be clarified. The lack of alveolar cell syncytia would be consistent with the lack of RSV infection. They present microscopic cross-sections the lung tissue and point to what they call syncytia, but it is not clear what they are seeing that leads them to that conclusion. More importantly, the sections are not stained for RSV antigens to determine if alveolar cells are infected by RSV and could therefore be responsible for the syncytia. They should be.

Ovine MSCs have been characterized before, but not lung resident MSCs (LR-MSCs) (l.136), which the investigators do here. They confirmed the multilineage capacity of the cells they isolated, a characteristic of MSCs. It is not clear why “ovine LR-MSCs transdifferentiated to chondrocytes, osteocytes, and adipocytes” would be relevant to repair of alveoli, but that is problem with the general concept of MSCs that is not unique to this report.

The investigators then examined pulmonary epithelial (CD31-CD45-panCTK+) and mesenchymal (CD31-CD45-panCTK-CD29+CD44+) compartments extracted from the lungs of lambs by bronchial epithelial lavage (BAL) and cultured. and detection of RSV in these cells. A micrograph in Fig. 4E has many yellow arrows pointing to a disturbance in the fibroblast monolayer which is not mentioned in the legend. It looks to be a syncytium. 4F shows a cluster of nuclei that are very close to each other and could also be a syncytium but that is also not mentioned in the legend. The size of the nuclei in the top (from uninfected) and bottom (from infected lambs) pictures is quite different but the size bars are the same.

Reviewer #2: This is a novel and timely study addressing the important question of the contribution of lung resident MSCs to the propagation and immune response to RSV infections. Using highly physiologically relevant models (pediatric human LR MSCs and lamb RSV infection model) authors, for the first time, demonstrated that MSCs are susceptable to RSV infections in vitro and in vivo. In vivo, analysis of transcriptional profile and CFU forming activity suggested that MSC play active role in modulation of anti-viral responses, lung repair and angiogenesis. Manuscript is well written, conslusions are supported by experimental evidence. I have no further questions and think that the manuscript can be published as is.

Reviewer #3: The authors reported a study on the response of lung-resident mesenchymal stem and stromal cells (LR-MSCs) to human RSV infection in a lamb model. They showed that primary pediatric LR-MSCs and LR-MSCs in the lamb model are permissive to RSV infection and also described the changes of their transcriptional profiles after RSV infection. The global transcriptional response of LR-MSCs was shown to follow RSV disease, switching from an early antiviral signature to repair mechanisms including differentiation, tissue remodeling, and angiogenesis. This is an interesting finding, but to make the massage clear, the following comments should be considered.

**Part II – Major Issues: Key Experiments Required for Acceptance**

Reviewer #1: The description of the RSV infection of in vitro-propagated MSCs must correlate with, rather than ignore the data. If syncytia are present, say so. If there is no virus produced and the infection does not spread o distant cells, don't claim that it does.

In addition to testing for all the proposed in vitro cell RSV receptors by RT-PCR, they need to test for the only in vivo receptor that has been described.

Provide evidence that RSV infects alveolar epithelial cells in vivo, or that it does not. An H&E stained section of alveoli described as a syncytium without a clear evidence that it is a syncytium is not enough. The syncytia must be clear and RSV antigen must be found in the cell.

Reviewer #2: (No Response)

Reviewer #3: 1. Fig. 1G and H: It was shown that there was almost no infectious RSV detectable in supernatants of infected LR-MSCs, while LR-MSCs are susceptible to RSV infection as shown in Fig. 1D. I think this finding is very interesting and important. For better understanding, show the RSV copies number in the supernatants in addition.

2. Fig. 2E and F: From these results, the authors concluded that RSV infection leads to a robust activation of LR-MSCs, characterized by a strong antiviral and pro-inflammatory phenotype combined with cytokines modulating T cell function. Concerning this, the involvement of the NS1 and NS2 proteins of RSV should be consider since these proteins are well known to be an IFN-antagonist inhibiting expression of antiviral host genes. It seems that the NSs function works well in WD-AECs but not in LR-MSCs.

3. Fig. 6E: What is the RSV positive rate in LR-MSCs at 6 days? Does the transcriptional profile represent for RSV-infected LR-MSCs? There is a dramatic change with 14 days from 6 days. Was the expression of the genes changed in the same population of LR-MSCs? Or were these different populations? It is possible that the majority of RSV-infected LR-MSCs died before 14 days and uninfected LR-MSCs were newly generated. Discuss about the fate of RSV-infected LR-MSCs.

**Part III – Minor Issues: Editorial and Data Presentation Modifications**

Reviewer #1: A more careful description and differentiation between airway epithelial cells and alveolar epithelial cells is needed. They are not the same, as is implied here (referencing literature on airway epithelial cells and claiming it support their idea of alveolar cell infection by RSV.

Reviewer #2: (No Response)

Reviewer #3: 4. References: Lines 348-351 (MSCs are applied in cell-based therapies with promising outcomes for the treatment of pulmonary morbidities such as idiopathic pulmonary fibrosis, acute-respiratory distress syndrome and severe influenza infections [10, 351 56-58]): Suggest citing here below article:

Yudhawati R, Amin M, Rantam FA, Prasetya RR, Dewantari JR, Nastri AM, Poetranto ED, Wulandari L, Lusida MI, Koesnowidagdo S, Soegiarto G, Shimizu YK, Mori Y, Shimizu K. Bone marrow-derived mesenchymal stem cells attenuate pulmonary inflammation and lung damage caused by highly pathogenic avian influenza A/H5N1 virus in BALB/c mice. BMC Infect Dis. 2020 Nov 11;20(1):823. doi: 10.1186/s12879-020-05525-2. PMID: 33176722; PMCID: PMC7656227.

They reported that the administration of MSCs prevented further lung injuries and inflammation caused by a highly pathogenic avian influenza A/H5N1 virus, and enhanced alveolar cell type II and I regeneration.

PLOS authors have the option to publish the peer review history of their article (what does this mean?). If published, this will include your full peer review and any attached files.

Reviewer #1: No

Reviewer #2: No

Reviewer #3: No
---

## [Editor Report · Decision Letter 1]

8 Jul 2021

Dear Dr. Alves,

We are pleased to inform you that your manuscript 'Pulmonary mesenchymal stem cells are engaged in distinct steps of host response to respiratory syncytial virus infection' has been provisionally accepted for publication in PLOS Pathogens.

Best regards,

Sabra L. Klein

Associate Editor

PLOS Pathogens

Carolina Lopez

Section Editor

PLOS Pathogens

Kasturi Haldar

Editor-in-Chief

PLOS Pathogens

orcid.org/0000-0001-5065-158X

Michael Malim

Editor-in-Chief

PLOS Pathogens

orcid.org/0000-0002-7699-2064

The authors thoughtfully addressed the concerns raised by Reviewers 1 and 3 pertaining to RSV infection and spread. The additional experimentation adequately addresses the major concerns.
---

## [Editor Report · Acceptance letter]

23 Jul 2021

Dear Dr. Alves,

We are delighted to inform you that your manuscript, "Pulmonary mesenchymal stem cells are engaged in distinct steps of host response to respiratory syncytial virus infection," has been formally accepted for publication in PLOS Pathogens.

Best regards,

Kasturi Haldar

Editor-in-Chief

PLOS Pathogens

orcid.org/0000-0001-5065-158X

Michael Malim

Editor-in-Chief

PLOS Pathogens

orcid.org/0000-0002-7699-2064